# KnowMol: Advancing Molecular Large Language Models with Multi-Level Chemical Knowledge

Zaifei Yang[1,2]    Hong Chang [1,2✉]    Ruibing Hou[1]    Shiguang Shan[1,2]    Xilin Chen[1,2]

[1]State Key Laboratory of AI Safety, Institute of Computing Technology, CAS, China
[2]University of Chinese Academy of Sciences (CAS), China
zyangea@connect.ust.hk,
{changhong, houruibing, sgshan, xlchen}@ict.ac.cn

## Abstract

The molecular large language models have garnered widespread attention due to their promising potential on molecular applications. However, current molecular large language models face significant limitations in understanding molecules due to inadequate textual descriptions and suboptimal molecular representation strategies during pretraining. To address these challenges, we introduce KnowMol-100K, a large-scale dataset with 100K fine-grained molecular annotations across multiple levels, bridging the gap between molecules and textual descriptions. Additionally, we propose chemically-informative molecular representation, effectively addressing limitations in existing molecular representation strategies. Building upon these innovations, we develop KnowMol, a state-of-the-art multi-modal molecular large language model. Extensive experiments demonstrate that KnowMol achieves superior performance across molecular understanding and generation tasks.

GitHub: `https://github.com/yzf-code/KnowMol`

Huggingface: `https://hf.co/datasets/yzf1102/KnowMol-100K`

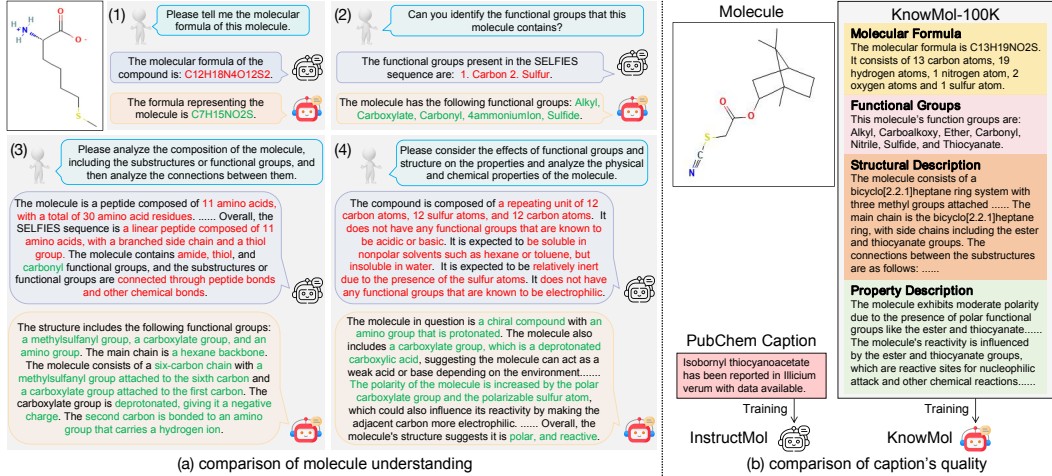

Figure 1: (a) Demonstration of InstructMol (baseline) and KnowMol (ours) on four fundamental molecular understanding factors: (1) atoms, (2) functional groups, (3) structure, and (4) properties. Error/hallucination parts are marked in red, while correct parts in green. InstructMol is the state-of-the-art among open-sourced Mol-LLMs. (b) The comparison between the caption in our KnowMol-100K and the widely used caption in PubChem database.

# 1 Introduction

The remarkable capabilities of large language models (LLMs) have spurred significant interest in developing molecular large language models (Mol-LLMs) [25, 32, 30, 15, 2, 3, 16]. These Mol-LLMs have demonstrated promising potential in tasks such as understanding individual molecular structures [30] and predicting chemical reactions involving multiple molecules [2, 16]. Despite the notable progress in molecule-related tasks, existing Mol-LLMs still fall short of achieving optimal comprehension of molecular information [39, 17]. As illustrated in Figure 1(a), our analysis discovers several limitations: Inaccurate identifications of molecular formulas and functional groups, imprecise interpretations of substructures, and wrong characterization of chemical connections and properties. These shortcomings underscore the capability of current Mol-LLMs on molecule understanding, thereby undermining their effectiveness in addressing complex chemistry tasks.

The main reasons for these problems lie in two aspects: *(i) low quality of pretraining dataset* and *(ii) sub-optimal molecular representation strategies*. On one hand, the PubChem database [19], commonly used for pretraining Mol-LLMs, exhibits two major deficiencies: imbalanced coverage and coarse granularity. Such descriptions fail to capture the complexity of molecular structures and properties, as shown in Figure 1(b). Existing works pay little attention to the improvement of the datasets' quality. Although HIGHT [3] attempts to address this problem by augmenting the captions with functional groups, it is far from enough for Mol-LLMs to understand and reason the structures and property details of molecules. On the other hand, existing molecular representation strategies, on one-dimensional (1D) string formats and two-dimensional (2D) graphical formats, may not effectively encode molecular information. For 1D representation, current methods [30, 16] usually utilize the SMILES [49] and apply the same tokenizer for both natural language and SMILES. However, SMILES suffer from inherent limitations [39], and the shared tokenization may lead to potential modality confusion for LLMs. For 2D representation, current approaches often employ a graph neural network [30, 2] or a specialized molecular tokenizer [3, 16] to encode molecule graphs. While these methods are effective for basic alignment, they fail to capture hierarchical structural information efficiently.

In this paper, we advance molecule large language models by tackling the above two challenges. To address the dataset challenge, we propose KnowMol-100K, the first comprehensive dataset with 100K multi-level molecule descriptions. Specifically, we design an elaborate pipeline with high-quality databases and tools to construct multi-level annotations from four fundamental factors: atoms, functional groups, molecular structures, and molecular properties. Consequently, the dataset increases both the coverage and granularity over PubChem captions, as shown in Figure 1(b).

Leveraging KnowMol-100K, we construct two instruction-following training tasks, including molecular understanding and generation, to enhance the capability of LLMs in understanding molecules.

To tackle the representation challenge, we introduce chemically-informative molecular representation strategies. For 1D strings, we replace SMILES with the more robust SELFIES [20] and design specialized vocabulary to avoid token-sharing issues with natural language. For 2D graphs, we propose an efficient hierarchical encoder that represents molecule graphs with multi-level tokens, capturing structural hierarchies without additional parameters.

Equipped with the sophisticated training tasks and the chemically-informative molecular representation strategies, we develop a state-of-the-art molecular large language model, KnowMol. Figure 2 shows the strong improvement of our model on 7 downstream tasks. KnowMol surpasses existing Mol-LLMs, including InstructMol [2] and HIGHT [3], in all tasks while making a clear advantage compared with UniMoT [16]. Qualita-

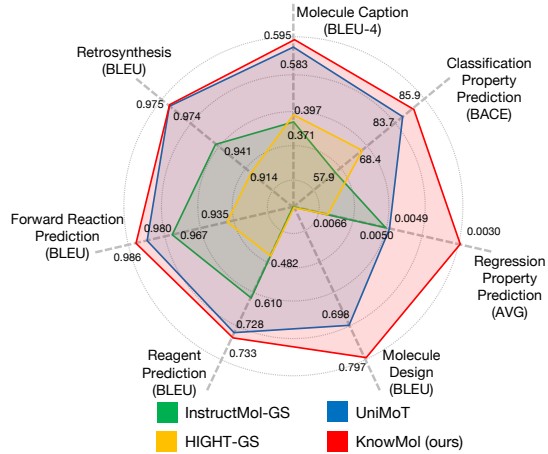

Figure 2: Our proposed KnowMol, a Mol-LLM with state-of-the-art performance on 7 molecule understanding and generation tasks.

tive analysis in Figure1(a) shows that KnowMol shows clear advantage on fundamental molecular understanding over the baseline, thus can be applied to a wider range of downstream molecular understanding and generation tasks.

In summary, this paper makes the following contributions:

- We discover the limitations of the pretraining datasets of Mol-LLMs and construct KnowMol-100K, which consists of 100K detailed multi-level molecular descriptions.

- We develop a chemically-informative molecular representation strategy using specialized tools on both 1D and 2D representations.

- Leveraging the elaborately crafted datasets and improved molecular representation strategies, we present KnowMol, a state-of-the-art Mol-LLM, which consistently outperforms existing models in various molecular understanding and generation tasks.

## 2 Related Work

**Molecule-text Data Enhancement by LLMs.** In the realm of molecule-text multi-modality, various methods have explored leveraging LLMs to enhance molecule-text data. Early method [53] uses MolT5 [10] to generate alternating dialogue data for CheBI-20 [9]. Benefiting from the rapid progress of GPT models, [24] utilized GPT-3.5 for semantic enrichment of sparse molecular descriptions in PubChem, while [42] employed GPT-4 to refine the construction of molecular caption data for instruction-based tasks. In addition, [4] applied few-shot prompting, using PubChem molecular annotations as examples, to generate an "artificially-real" dataset with ChatGPT for domain adaptation. Another approach [11] combined multiple datasets with GPT-4 to construct templates and integrate them with original data to create diversified molecular descriptions. Despite these diverse efforts, all the aforementioned methods rely on PubChem as their primary data source, which inherently limits the quality of the generated captions due to the original data shortcomings. In contrast, our approach utilizes advanced tools to construct a multi-level, fine-grained molecule-text dataset, overcoming the limitations within PubChem descriptions.

**Molecule Graph Representation Learning for LLMs.** To enable LLMs to handle molecule graphs, several methods have been proposed to achieve informative graph representations. Early models [43, 28, 31, 26] employ GNNs as molecular encoders and utilize cross-modal contrastive learning to align molecular and textual representation spaces. Subsequently, multi-modal architectures incorporating adapter-based mechanisms with LLMs have been explored. For example, models such as InstructMol [2] and DrugChat [25] integrate simple projection layers to map molecular features into the LLM input space, while architectures like MolCA [30] and 3D-MoLM [24] leverage Q-Former [23] modules to bridge modality gaps. Recently, recognizing the limitations of existing molecular representation approaches, HIGHT [3] and UniMoT [16] have proposed specially designed tokenizers to enhance the quality of molecular representations. However, their approach employs complicated models, such as Vector Quantized Variational AutoEncoders (VQ-VAEs) [47] or Q-former [23], necessitating an additional pretraining stage and significantly increasing computational complexity. Despite various attempts in model designs, a key limitation persists: how to improve molecular representation in both 1D and 2D modalities which is efficient and effective?

## 3 KnowMol-100K Dataset

### 3.1 Preliminaries

Several fundamental factors are essential for a comprehensive understanding of molecular characteristics [41, 35]: **(a) Atoms and functional groups**, as the fundamental units of molecular structure, which serve as the primary interaction sites and determine a molecule's core composition and inherent properties. **(b) Molecular structure**, which defines the arrangement and bonding of atoms and functional groups, and governs the geometry and spatial configuration of molecules. **(c) Physicochemical properties**, including six important aspects [33, 34, 7, 38, 48, 37]: polarity, acidity/basicity, solubility, reactivity, stereochemistry, and electrophilicity. These properties are influenced by the atomic composition, functional groups, and overall structure, and play a crucial role in determining the behavior and interactions of molecules in diverse environments, thereby influencing their applications

Table 1: Statistics of two subsets of 1000 PubChem descriptions on the coverage and granularity of fundamental factors for molecule understanding. The number outside (in) parentheses indicates the average word count (amount of occurrence) of the fundamental factor.

| sample set | atoms | Functional groups | molecular structure | Polarity | Acidity/Basicity | Physicochemical property | | | | Electrophilicity | Sum | full description |
|---|---|---|---|---|---|---|---|---|---|---|---|---|
| | | | | | | Solubility | Reactivity | Stereochemistry | | | | |
| random sampling | 1.643 (104) | 2.454 (259) | 2.406 (165) | 0.028 (4) | 0.566 (51) | 0.114 (16) | 1.019 (38) | 0.304 (29) | 0.035 (1) | 2.066 (139) | 19.338 |
| long text sampling | 8.428 (404) | 9.579 (660) | 14.178 (656) | 0.145 (17) | 2.572 (184) | 0.892 (59) | 9.965 (317) | 0.885 (69) | 0.011 (1) | 14.470 (647) | 68.283 |

in chemical processes. Together, these factors underpin molecular behavior and are indispensable for accurate molecular description.

Based on these fundamental factors, we first describe the shortages of the existing dataset, PubChem, on these factors in Sec.3.2, highlighting the necessity of constructing KnowMol-100K. Then we describe the construction of KnowMol-100K targeting these factors, from the perspective of data sampling and annotation pipeline in Sec.3.3.

## 3.2 Shortcomings of Existing Dataset

In this section, we provide an in-depth analysis of the critical molecular information present (or absent) in the PubChem dataset.

Existing molecular deep learning methods [30, 2, 16] primarily rely on the PubChem database to construct the molecule–caption pair datasets. To assess the coverage of fundamental molecular characteristics in the PubChem dataset, Table 1 summarizes detailed statistics of the aforementioned factors from two subsets of PubChem descriptions. Specifically, we construct subsets of size 1000 using two strategies: (a) random selection from all of the descriptions, and (b) random selection from descriptions longer than 40 words. We leverage GPT-4 to select the content related to each factor, and then calculate the average length (by word) of relevant content in each sample. This provides a reference metric for how thoroughly each factor is covered.

**Observation 1: PubChem Captions Cover Only a Limited Subset of Aspects.** As depicted in Table 1, we observe a pronounced imbalance in the average word count and the occurrence across different factors. Polarity and electrophilicity, for instance, appear nearly absent, whereas certain details such as functional groups and reactivity are covered in relatively more detail. This imbalance suggests that PubChem captions focus disproportionately on a few aspects while overlooking other critical components like solvent-related properties or stereochemical nuances. Such selective coverage restricts the depth and breadth of molecular understanding of Mol-LLMs pretrained on these captions.

**Observation 2: PubChem Captions Provide Only Coarse-grained Annotations.** Another key limitation of PubChem captions lies in their brevity. As shown in Table 1, all of the descriptions exhibit a relatively low average word count and insufficient occurrence for crucial molecular aspects. For example, while captions may briefly mention the presence of functional groups, the explanations rarely extend to discuss how these groups connect or contribute to the overall property. Similarly, references to specific properties tend to be perfunctory, omitting critical nuances that would substantially enrich a model's comprehension ability. Consequently, such shallow, sparse, and coarse-grained captions limit the development of Mol-LLMs for representing and understanding the complexity of molecular structure and properties.

## 3.3 Construction Pipeline

### 3.3.1 Data Sampling

PubChem database records a large amount of basic information about molecules, such as molecular formula, IUPAC name, and molecule picture. To leverage these resources, we choose molecules with the basic information available from the PubChem database. However, given the large size of the molecules in the PubChem database and the high similarity among molecules, annotating the dataset directly would result in significant redundancy. To address this problem, we implement a screening process to reduce the molecule set's size and redundancy. In detail, we select a subset of 100,000 molecules exhibiting maximal diversity using the MaxMin method [1] to maximize the diversity of the molecules in the dataset. This selected molecule set is used for subsequent annotation.

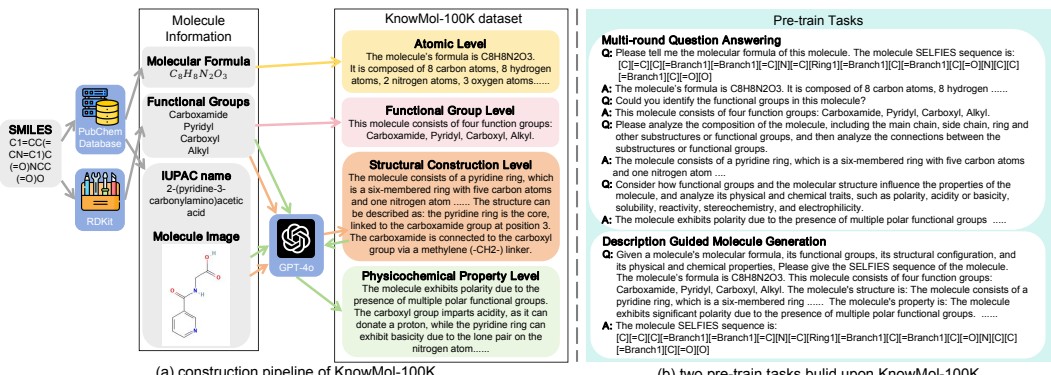

Figure 3: (a) The pipeline of building the KnowMol-100K. We use a combination of basic data from PubChem databases, an open-source toolkit for cheminformatics, RDKit, and the powerful multi-modal large language model GPT-4o. (b) Building upon the KnowMol-100K, we design two instruction-following pre-train tasks: (1) Multi-round Question Answering, and (2) Description Guided Molecule Generation.

### 3.3.2 Multi-level Annotations

Based on the multi-level molecular structure and the dependencies between levels as introduced in Sec 3.1, we developed a multi-level, fine-grained dataset called KnowMol-100K. The construction process of this dataset integrates the basic information of the PubChem database, the functional group analysis results of the cheminformatics toolkit RDKit [21], and the detailed language description generated by GPT-4o[36]. The annotations are divided into four levels of chemical knowledge, organized from basic to complex: (1) atomic level, (2) functional group level, (3) structural construction level, and (4) physicochemical property level. The dataset construction pipeline is illustrated in Figure 3(a). Next, we delve into the construction of the four levels of annotations. For more details, please see Appendix A.

- **Atomic Level.** At this level of annotation, we leverage the molecular formula data from the PubChem database. By parsing the chemical formula, we identify the types of atoms and their corresponding quantities that constitute the molecule.

- **Functional Group Level.** This level of annotation utilizes the chemical informatics toolkit RDKit, and a collection of patterns for 82 common functional groups built by [14]. RDKit is used to identify the matched functional groups within the molecule based on the Breaking of Retrosynthetically Interesting Chemical Substructures (BRICS) algorithm [8]. Notably, the BRICS algorithm is a deterministic matching process, which is highly reliable for this task and could guarantee the correctness of Functional Group annotations.

- **Structural Construction Level.** At this level of annotation, we leverage SMILES formulas, IUPAC names, molecule images from the PubChem database, and functional group annotations generated by the previous level. All molecules in our datasets are equipped with the above basic information. By incorporating basic information from different sources and aspects, GPT-4o is prompted to analyze the relationships between the main chain, side chains, rings, and their associated functional groups accurately and efficiently, thereby creating a detailed description of the molecular structure. This multifaceted information can provide comprehensive basic knowledge of the molecular structure from various complementary perspectives, thus ensuring the consistency and precision of GPT-4o in generating accurate descriptions.

- **Physicochemical Property Level.** To annotate the physicochemical property level, we leverage the above annotated functional group and structural construction to prompt GPT-4o to analyze the physicochemical properties derived from functional groups and their interactions within the molecular structure. Along with the rich information, we also prompt GPT-4o with explicit definitions of six specific properties: polarity, acidity/basicity, solubility, reactivity, stereochemistry, and electrophilicity, ensuring the reliability of GPT-4o in generating accurate and complete descriptions.

### 3.3.3 Dataset Quality Inspection

To evaluate the quality of KnowMol's annotations, we invite three chemical expert volunteers from a national chemical research institute to conduct qualitative evaluations from multiple perspectives.

The evaluation results show the strong reliability and quality of our dataset. The detailed evaluation process and results can be found in the appendix B.

## 4 Chemically-Informative Molecular Representation Learning

Following the implementation of InsturctMol [2], our baseline model consists of three components: (1) a molecule graph encoder, (2) a projection layer, and (3) a LLM. Based on the baseline model, we make improvements on molecular representation learning on both molecule string and molecule graph, as presented in Sec.4.1 and Sec.4.2 respectively. The chemically-informative model architecture is illustrated in Figure 4.

### 4.1 Molecule Tokenization for String

SMILES [49] is a widely used string representation for molecules, but several studies [39, 15] have highlighted its inherent limitations. In response to these concerns, we adopted the improved SELFIES [20] representation for molecules. Furthermore, some studies [39, 45] have raised doubts about the efficacy of sharing token embeddings across molecule string and natural language, leading to the design of separate vocabularies for each modality. In alignment with these findings, we constructed a dedicated token vocabulary for SELFIES, where each chemically meaningful atom group, denoted by brackets in the SELFIES syntax, is treated as a distinct token. This approach ensures that the semantic spaces of different modalities remain clearly separated, preserving the integrity of each modality and preventing potential cross-modal confusion.

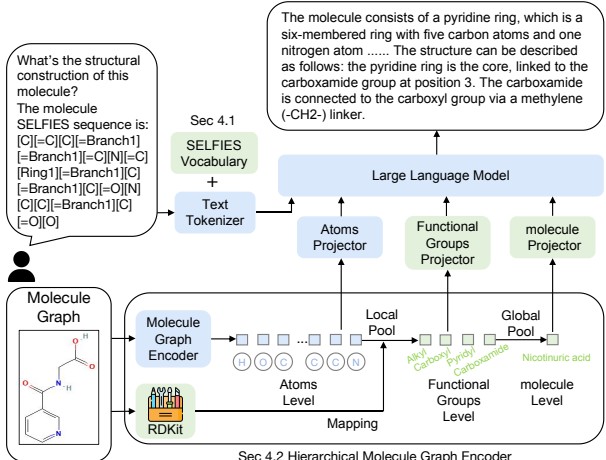

Figure 4: Chemically-informative model architecture. Three levels of representation are used for molecular features: atomic level, functional group level, and molecule level. We mark baseline model in blue, while improved parts in green.

### 4.2 Hierarchical Tokenization for Molecule Graph

Current Mol-LLMs [15, 2] normally use the graph neural network as the encoder only to extract atomic-level molecular tokens. However, atomic-level tokens alone are inadequate for capturing the inherent hierarchical structure of molecules. Drawing inspiration from multi-level molecular annotations, the use of hierarchical tokens for representing molecule graphs offers a more informative and nuanced approach. Specifically, the incorporation of functional group tokens and molecule tokens serves to encapsulate higher-order structural and chemical information, enabling more effective communication of molecular structures and properties to LLMs. By leveraging these hierarchical tokens, the graph-language alignment with the multi-level annotations annotated in KnowMol-100K will be significantly enhanced. This approach could facilitate more accurate and context-aware reasoning about molecular data, improving the model's ability to understand and generate meaningful interpretations of complex chemical graphs.

To achieve this, we design an efficient graph hierarchical encoder. Same as the functional group level annotation in KnowMol-100K, we use the RDKit [21] and BRICS algorithm [8] to detect the functional group in the molecule and get the mapping between functional groups and their constituent atoms. Using the obtained mapping, we employ local pooling on the corresponding atoms within a functional group to get functional group level tokens. Subsequently, these functional group-level tokens are further globally pooled to form molecule-level tokens. Outside the encoder, the three level tokens are further projected using separate projectors into the LLM's embedding space. Using the BRICS algorithm and pooling, we constructed molecular tokens with hierarchical dependencies,

transforming the original atomic-level tokens into a more detailed and hierarchical representation, without bringing additional training parameters or extra model usage in the encoder.

## 5    Experiments

### 5.1    Implementation

**Construct Pre-train Tasks from KnowMol-100K.** Building upon the four levels of molecular knowledge annotated in KnowMol-100K, we utilize molecular information from each annotation level to design two instruction-following pre-train tasks. Figure 3(b) illustrates the pre-train tasks.

- **Multi-Round Question Answering.** The first task involves a multi-round, iterative question-answering process. The questions commence with fundamental atomic information and progressively advance to more complex topics, such as functional groups, molecular structures, and physicochemical properties.
- **Description Guided Molecule Generation.** The second task requires the model to generate the corresponding molecule based on the four-level annotations, presenting a reverse challenge to the first task. Through training on this task, the model learns to generate molecules grounded in specified molecular structures or chemical properties.

**Training Setting.** Based on the two pre-train tasks, we trained our model, KnowMol. The training of KnowMol is divided into two instruction-tuning stages:

- **Pretraining.** The pretraining stage uses the tasks constructed in Sec 5.1 to inject comprehensive chemical knowledge into the LLM. Given these high-quality data, only fine-tuning the projection layers does not suffice to exploit the full capabilities. So this stage involves fine-tuning LLM using low-rank adaptation (LoRA) [18] and the projection layers. The molecule graph encoder is frozen to avoid feature interference.
- **Task-specific Instruction Tunning.** The second stage fine-tunes KnowMol for specific downstream tasks, allowing it to effectively interpret and follow human instructions, thereby enhancing the model's performance across various applications. We also utilize LoRA to improve efficiency.

### 5.2    Molecule Comprehension Tasks

**Baselines.**  Following previous work [2, 3], We adapt three types of baselines: (1) **Specialist Models**, (2) **Retrieval Based LLM**, and (3) **LLM-Based Generalist Models**. Specialist Models refer to single-modality molecular models that are pre-trained on large molecular datasets using either supervised or unsupervised tasks and then fine-tuned on specific downstream tasks. Retrieval Based LLM approaches mainly utilize ChatGPT or GPT-4 as the foundation model and employ retrieval methods on the molecule captioning task. LLM-Based Generalist Models include base large language models and other models built on top of the LLMs via instruction tuning or architectural improvements. These generalist models have open-form communication capabilities and can be flexibly adapted to various specialized tasks by switching different adapters.

As investigated by [40], generalist models are not expected to outperform specialist models universally. We follow the prior works [2, 3] to both highlight the best Specialist Models and LLM-Based Generalist Models. In general, KnowMol demonstrates clear and robust improvements over LLM-based generalist models, which are our primary baselines. Besides, KnowMol also performs comparably to Specialist Models while having broad versatility across multiple tasks.

**Molecule Captioning Task.** The molecule captioning task requires the model to generate the given molecule's description. We conduct the experiment on the widely used dataset ChEBI-20 [9] in instruction tuning format. Since UniMoT [16] uses data from PubChem pre-train set in their Pre-training stage with LoRA, in order to compare with it fairly, we also report the result of fine-tuning our model on PubChem pre-train set without duplication with the ChEBI-20 test split.

The results are listed in Table 2. We can observe consistent improvements above the baselines across multiple evaluation metrics(BLEU, ROUGE-2, ROUGE-L). Compared to specialist models such as Text+Chem T5-augm-base, which previously held the best results among baselines, Know-Mol surpasses it by 0.053 on BLEU-4 and 0.058 on ROUGE-2. When compared to generalist

Table 2: Results of the molecular description generation task on the test split of the ChEBI-20 dataset.

| MODEL | BLEU-2↑ | BLEU-4↑ | ROUGE-1↑ | ROUGE-2↑ | ROUGE-L↑ | METEOR↑ |
|---|---|---|---|---|---|---|
| *Specialist Models* | | | | | | |
| MoT5-base [10] | 0.540 | 0.457 | 0.634 | 0.485 | 0.568 | 0.569 |
| MoMu (MolT5-base) [43] | 0.549 | 0.462 | - | - | - | 0.576 |
| MolFM (MolT5-base) [31] | 0.585 | 0.498 | 0.653 | 0.508 | 0.594 | 0.607 |
| MolXPT [29] | 0.594 | 0.505 | 0.660 | 0.511 | 0.597 | 0.626 |
| GIT-Mol-graph [26] | 0.290 | 0.210 | 0.540 | 0.445 | 0.512 | 0.491 |
| GIT-Mol-SMILES [26] | 0.264 | 0.176 | 0.477 | 0.374 | 0.451 | 0.430 |
| GIT-Mol-(graph+SMILES) [26] | 0.352 | 0.263 | 0.575 | 0.485 | 0.560 | 0.430 |
| Text+Chem T5-augm-base [6] | **0.625** | **0.542** | **0.682** | **0.543** | **0.622** | **0.648** |
| *Retrieval Based LLMs* | | | | | | |
| GPT-3.5-turbo (10-shot MolReGPT) [22] | 0.565 | 0.482 | 0.623 | 0.450 | 0.543 | 0.585 |
| GPT-4-0314 (10-shot MolReGPT) [22] | 0.607 | 0.525 | 0.634 | 0.476 | 0.562 | 0.610 |
| *LLM Based Generalist Models* | | | | | | |
| GPT-3.5-turbo (zero-shot) [22] | 0.103 | 0.050 | 0.261 | 0.088 | 0.204 | 0.161 |
| BioMedGPT-10B [32] | 0.234 | 0.141 | 0.386 | 0.206 | 0.332 | 0.308 |
| Mol-Instructions [15] | 0.249 | 0.171 | 0.331 | 0.203 | 0.289 | 0.271 |
| InstructMol-GS [2] | 0.475 | 0.371 | 0.566 | 0.394 | 0.502 | 0.509 |
| HIGHT-GS [3] | 0.498 | 0.397 | 0.582 | 0.414 | 0.518 | 0.525 |
| MolCA [30] | 0.620 | 0.531 | 0.681 | 0.537 | 0.618 | 0.651 |
| UniMoT [16] | 0.664 | 0.583 | **0.722** | 0.584 | 0.664 | **0.703** |
| KnowMol (finetuned on ChEBI-20) | 0.605 | 0.518 | 0.666 | 0.522 | 0.605 | 0.626 |
| KnowMol (finetuned on ChEBI-20 and PubChem pretrain-set) | **0.665** | **0.595** | 0.717 | **0.601** | **0.671** | 0.683 |

Table 3: Molecular property prediction task (classification) on the MoleculeNet benchmark. We report the ROC-AUC metric for classification tasks. *: Fine-tuned with LoRA.

| METHOD # MOLECULES | BACE↑ 1513 | BBBP↑ 2039 | HIV↑ 41127 | MUV↑ 93087 | Tox21↑ 7831 |
|---|---|---|---|---|---|
| *Specialist Models* | | | | | |
| KV-PLM [52] | 78.5 | 70.5 | 71.8 | 61.7 | 49.2 |
| GraphMVP-C [27] | 81.2 | 72.4 | 77.0 | 74.4 | 77.1 |
| MoMu [43] | 76.7 | 70.5 | 75.9 | 60.5 | 57.8 |
| MolFM [31] | 83.9 | **72.9** | 78.8 | 76.0 | 77.2 |
| Uni-Mol [55] | **85.7** | **72.9** | 80.8 | **82.1** | 78.1 |
| GIMLET [54] | 69.6 | 59.4 | 66.2 | 64.4 | 61.2 |
| *LLM Based Generalist Models* | | | | | |
| Galactica-6.7B [45] | 58.4 | 53.5 | 72.2 | - | 63.9 |
| Galactica-30B [45] | 72.7 | 59.6 | 75.9 | - | 68.5 |
| Galactica-120B [45] | 61.7 | 66.1 | 74.5 | - | 68.9 |
| Vicuna-v1.5-13b-16k (4-shot) [5] | 49.2 | 52.7 | 50.5 | - | - |
| Vicuna-v1.3-7b* [5] | 68.3 | 60.1 | 58.1 | - | - |
| LLama-2-7b-chat* [13] | 74.8 | 65.6 | 62.3 | 46.9 | 62.0 |
| InstructMol-G [2] | 64.3 | 48.7 | 50.2 | 50.0 | 59.0 |
| HIGHT-GS [3] | 77.1 | 61.8 | 63.3 | 51.1 | 67.4 |
| KnowMol | **85.9** | 69.2 | 81.8 | 61.5 | 68.7 |

Table 4: Results on molecular property prediction tasks (regression) on QM9. We report the MAE results of the hartree metric. $\Delta\epsilon$: HOMO-LUMO energy gap. †: few-shot in-context learning(ICL) results from Mol-Instructions. Baseline results are from Instructmol.

| METHOD | HOMO↓ | LUMO↓ | $\Delta\epsilon$↓ | AVG↓ |
|---|---|---|---|---|
| *LLM Based Generalist Models* | | | | |
| Alpaca† [44] | - | - | - | 322.109 |
| Baize† [51] | - | - | - | 261.343 |
| LLama2-7B [13] (5-shot ICL) | 0.7367 | 0.8641 | 0.5152 | 0.7510 |
| Vicuna-13B [5] (5-shot ICL) | 0.7135 | 3.6807 | 1.5407 | 1.9783 |
| Mol-Instructions [15] | 0.0210 | 0.0210 | 0.0203 | 0.0210 |
| InstructMol-GS [2] | 0.0048 | 0.0050 | 0.0061 | 0.0050 |
| HIGHT-GS [3] | 0.0056 | 0.0065 | 0.0077 | 0.0066 |
| UniMoT [16] | 0.0042 | 0.0047 | 0.0055 | 0.0049 |
| KnowMol | **0.0028** | **0.0029** | **0.0034** | **0.0030** |

models, KnowMol also achieves substantial improvements. This strong performance highlights KnowMol's advanced capability in generating molecular descriptions, showcasing the effectiveness of our enhanced dataset and model architecture.

**Molecule Property Prediction Task.** The Molecule Property Prediction Task requires the model to predict the molecule's specific property. We leverage 5 classification tasks(BACE, BBBP, HIV, MUV, Tox21) from MoleculeNet [50] with the standard scaffold splitting and the instruction tuning formats from GIMLET [54]. We also leverage the regression tasks built on the QM9 dataset by Mol-Instructions [15]. For classification tasks and regression tasks, we report the ROC-AUC metric and the Mean Absolute Error (MAE) metric respectively.

Compared to the baselines, KnowMol performs strong advance in both Table 3 and 4, indicating its better understanding of molecular properties, demonstrating the significant effectiveness of our dataset and representation strategies in bridging the basic molecule string and the complex molecule properties. This effectiveness highlights the potential of KnowMol for more complex tasks related to molecular properties.

## 5.3 Molecule Generation Tasks

For molecule generation tasks, we choose LLM-Based Generalist Models as the baselines and incorporate four datasets from [15] i.e., caption-guided molecule generation, reagent prediction, forward reaction prediction, and retrosynthesis prediction. Caption-guided molecule generation aims to generate the corresponding molecule of the given description. Reagent prediction aims to determine the catalysts, solvents, or ancillary substances required for a specific chemical reaction based on the given reactant(s) and product(s). Forward reaction prediction aims to predict the possible products

Table 5: Results of molecule generation tasks. †: Few-shot ICL results from Mol-Instructions. ∗: fine-tuned using task-specific instruction data.

| MODEL | EXACT↑ | BLEU↑ | LEVENSHTEIN↓ | RDK FTS↑ | MACCS FTS↑ | MORGAN FTS↑ | VALIDITY↑ |
|---|---|---|---|---|---|---|---|
| *Caption-guided Molecule Generation* | | | | | | | |
| LLama [46] | 0.000 | 0.003 | 59.864 | 0.005 | 0.000 | 0.000 | 0.003 |
| Vicuna [5] | 0.000 | 0.006 | 60.356 | 0.006 | 0.001 | 0.000 | 0.001 |
| Mol-Instructions [15] | 0.002 | 0.345 | 41.367 | 0.231 | 0.412 | 0.147 | 1.000 |
| MolT5 [46](LoRA) | 0.112 | 0.546 | 38.276 | 0.400 | 0.538 | 0.295 | 0.773 |
| UniMoT [16] | **0.237** | 0.698 | **27.782** | 0.543 | 0.651 | 0.411 | 1.000 |
| KnowMol | 0.083 | **0.797** | 30.702 | **0.570** | **0.693** | **0.426** | 1.000 |
| *Reagent Prediction* | | | | | | | |
| Alpaca[†] [44] | 0.000 | 0.026 | 29.037 | 0.029 | 0.016 | 0.001 | 0.186 |
| Baize[†] [51] | 0.000 | 0.051 | 30.628 | 0.022 | 0.018 | 0.004 | 0.099 |
| ChatGLM[†] [12] | 0.000 | 0.019 | 29.169 | 0.017 | 0.006 | 0.002 | 0.074 |
| LLama[†] [46] | 0.000 | 0.003 | 28.040 | 0.037 | 0.001 | 0.001 | 0.001 |
| Vicuna[†] [5] | 0.000 | 0.010 | 27.948 | 0.038 | 0.002 | 0.001 | 0.007 |
| Mol-Instructions [15] | 0.044 | 0.224 | 23.167 | 0.237 | 0.364 | 0.213 | 1.000 |
| LLama-7b∗ [46](LoRA) | 0.000 | 0.283 | 53.510 | 0.136 | 0.294 | 0.106 | 1.000 |
| InstructMol-GS [2] | 0.129 | 0.610 | 19.664 | 0.444 | 0.539 | 0.400 | 1.000 |
| HIGHT-GS [3] | 0.067 | 0.482 | 27.167 | 0.462 | 0.346 | 0.303 | 1.000 |
| UniMoT [16] | 0.167 | 0.728 | 14.588 | **0.549** | **0.621** | **0.507** | 1.000 |
| KnowMol | **0.238** | **0.733** | **14.058** | 0.525 | 0.609 | 0.490 | 1.000 |
| *Forward Reaction Prediction* | | | | | | | |
| Alpaca[†] [44] | 0.000 | 0.065 | 41.989 | 0.004 | 0.024 | 0.008 | 0.138 |
| Baize[†] [51] | 0.000 | 0.044 | 41.500 | 0.004 | 0.025 | 0.009 | 0.097 |
| ChatGLM[†] [12] | 0.000 | 0.183 | 40.008 | 0.050 | 0.100 | 0.044 | 0.108 |
| LLama[†] [46] | 0.000 | 0.020 | 42.002 | 0.001 | 0.002 | 0.001 | 0.039 |
| Vicuna[†] [5] | 0.000 | 0.057 | 41.690 | 0.007 | 0.016 | 0.006 | 0.059 |
| Mol-Instructions [15] | 0.045 | 0.654 | 27.262 | 0.313 | 0.509 | 0.262 | 1.000 |
| LLama-7b∗ [46](LoRA) | 0.012 | 0.804 | 29.947 | 0.499 | 0.649 | 0.407 | 1.000 |
| InstructMol-GS [2] | 0.536 | 0.967 | 10.851 | 0.776 | 0.878 | 0.741 | 1.000 |
| HIGHT-GS [3] | 0.293 | 0.935 | 16.687 | 0.774 | 0.618 | 0.566 | 1.000 |
| UniMoT [16] | 0.611 | 0.980 | 8.297 | 0.836 | 0.911 | 0.807 | 1.000 |
| KnowMol | **0.752** | **0.986** | **5.662** | **0.889** | **0.943** | **0.872** | 1.000 |
| *Retrosynthesis* | | | | | | | |
| Alpaca[†] [44] | 0.000 | 0.063 | 46.915 | 0.005 | 0.023 | 0.007 | 0.160 |
| Baize[†] [51] | 0.000 | 0.095 | 44.714 | 0.025 | 0.050 | 0.023 | 0.112 |
| ChatGLM[†] [12] | 0.000 | 0.117 | 48.365 | 0.056 | 0.075 | 0.043 | 0.046 |
| LLama[†] [46] | 0.000 | 0.036 | 46.844 | 0.018 | 0.029 | 0.017 | 0.010 |
| Vicuna[†] [5] | 0.000 | 0.057 | 46.877 | 0.025 | 0.030 | 0.021 | 0.017 |
| Mol-Instructions [15] | 0.009 | 0.705 | 31.227 | 0.283 | 0.487 | 0.230 | 1.000 |
| LLama-7b∗ [46](LoRA) | 0.000 | 0.283 | 53.510 | 0.136 | 0.294 | 0.106 | 1.000 |
| InstructMol-GS [2] | 0.407 | 0.941 | 13.967 | 0.753 | 0.852 | 0.714 | 1.000 |
| HIGHT-GS [3] | 0.202 | 0.914 | 20.194 | 0.772 | 0.623 | 0.577 | 0.999 |
| UniMoT [16] | 0.478 | 0.974 | 11.634 | 0.810 | 0.909 | 0.771 | 1.000 |
| KnowMol | **0.598** | **0.975** | **8.363** | **0.856** | **0.912** | **0.829** | 1.000 |

given the reactant(s) and reagent(s). Retrosynthesis prediction aims to predict the potential reactant(s) given the product(s). These tasks evaluate the ability of LLMs to generate specific molecules based on the given conditions. The metrics evaluate the similarity between the generated molecule and the ground truth molecule from diverse aspects.

Table 5 shows the result of the molecule generation tasks. The evaluation across the four molecule generation tasks collectively highlights the versatility, accuracy, and chemical validity of KnowMol. Compared to other models, KnowMol demonstrates a balanced and comprehensive performance across all aspects of molecular generation, from semantic alignment with descriptions to structural accuracy and chemical plausibility. Its consistent superiority on metrics like BLEU, fingerprint similarity (RDK, MACCS, and Morgan), and Exact Match reflects its ability to capture both the textual and structural intricacies of molecular design. These advantages highlight that KnowMol's advanced dataset and model architecture enable it to not only outperform existing baselines in specific tasks but also maintain high performance across diverse molecular generation scenarios.

## 5.4 Ablation Study

Since accurate chemical reaction prediction requires the model's comprehensive understanding of all of the involved molecules across multiple perspectives, we choose forward reaction prediction as the ablation task. We perform LoRA tuning of LLM on the pre-training set of PubChem dataset[19] and take it as the baseline.

Table 6: Ablation study on the impact of annotation level, training task, and representation learning. SC: Structural Construction. PP: Physicochemical Property. MRQA: multi-round question answering. DGMG: description guided molecule generation. HR: hierarchical representation. ST: SELFIES tokenization.

| Training Data | HR | ST | Exact↑ | BLEU↑ | Levenshtein↓ | RDK FTS↑ | MACCS FTS↑ | Morgan FTS↑ | Validity↑ |
|---|---|---|---|---|---|---|---|---|---|
| PubChem Captions (301K) | ✗ | ✗ | 0.509 | 0.954 | 11.33 | 0.762 | 0.868 | 0.730 | 1.000 |
| MRQA w/o SC (100K) | ✗ | ✗ | 0.618 | 0.969 | 8.982 | 0.822 | 0.907 | 0.792 | 1.000 |
| MRQA w/o PP (100K) | ✗ | ✗ | 0.587 | 0.970 | 9.580 | 0.812 | 0.897 | 0.778 | 1.000 |
| MRQA (100K) | ✗ | ✗ | 0.627 | 0.967 | 9.250 | 0.827 | 0.905 | 0.795 | 1.000 |
| DGMG (100K) | ✗ | ✗ | 0.624 | 0.966 | 8.756 | 0.830 | 0.906 | 0.794 | 1.000 |
| MRQA + DGMG (200K) | ✗ | ✗ | 0.622 | 0.974 | 8.677 | 0.833 | 0.910 | 0.800 | 1.000 |
| MRQA + DGMG (200K) | ✓ | ✗ | 0.728 | 0.985 | 6.429 | 0.879 | 0.936 | 0.857 | 1.000 |
| MRQA + DGMG (200K) | ✓ | ✓ | **0.752** | **0.986** | **5.662** | **0.889** | **0.943** | **0.872** | 1.000 |

**Impact of Annotation Level.** We conduct ablation on the Structural Construction Level and Physicochemical Property Level annotation in the multi-round question answering training task. The comparison of lines 1-4 in Table 6 shows the indispensable role of both annotation levels. Serving as one part of the fundamental factors, both levels provide certain knowledge for Mol-LLMs and it is irreplaceable.

**Impact of Training Tasks.** We conduct ablation on the impact of two training tasks constructed using the KnowMol-100K, the result is shown in Table 6 lines 4-6. Excluding either training task would lead to a clear drop in performance, implying the necessity of both the molecule understanding task and the molecule generation task. Compared to the baseline trained on PubChem, our training data constructed based on KnowMol-100K shows significant improvement with less training data, reflecting the clear advantage of our dataset in efficiently enhancing the ability of Mol-LLMs.

**Impact of Representation Learning.** We conduct ablation to assess the influence of the enhanced molecular representation learning, the result is shown in Table 6 lines 6-8. The row 6 of Table 6 corresponds to the InstructMol [2] fine-tuned on KnowMol-100K with both training task, but without the enhanced representation strategies proposed in our work. Lines 7 and 8 show that enhancing either the 1D molecule string or 2D molecule graph representation leads to clear improvements, while combining them yields better performance. This indicates the effectiveness of both SELFIES tokenization and hierarchical molecule graph representation in enhancing the quality of molecular representation for Mol-LLMs.

# 6 Conclusion

In this work, we shed light on the untapped limitation of Mol-LLMs in fundamental molecule understanding and address critical challenges in Mol-LLMs, including the inadequacy of textual molecular descriptions and suboptimal representation strategies. To address these challenges, We introduce KnowMol-100K, a large-scale dataset with 100K multi-level annotations that bridges the gap between molecular information and textual descriptions, enabling a deeper understanding of molecules. Additionally, we propose chemically-informative molecular representation strategies that effectively capture the diversity and hierarchical structure of molecules. Building upon these contributions, we develop KnowMol. Extensive evaluations demonstrate that KnowMol significantly outperforms existing models in molecular understanding and generation tasks, highlighting its capability to address complex molecular challenges.

Overall, this study provides a foundation for advancing Mol-LLMs and highlights the potential of large-scale datasets and tailored representation strategies in bridging the gap between molecular science and artificial intelligence. While our contributions represent a clear step forward, there are limitations for further refinement. Future work could explore the usage of KnowMol to generate more high-quality data, as well as consider the integration of advanced three-dimensional molecular representations.

# Acknowledgments

This work is partially supported by the program of The Robotic AI-Scientist Platform of Chinese Academy of Sciences.

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

# NeurIPS Paper Checklist

The checklist is designed to encourage best practices for responsible machine learning research, addressing issues of reproducibility, transparency, research ethics, and societal impact. Do not remove the checklist: **The papers not including the checklist will be desk rejected.** The checklist should follow the references and follow the (optional) supplemental material. The checklist does NOT count towards the page limit.

Please read the checklist guidelines carefully for information on how to answer these questions. For each question in the checklist:

- You should answer [Yes] , [No] , or [NA] .
- [NA] means either that the question is Not Applicable for that particular paper or the relevant information is Not Available.
- Please provide a short (1–2 sentence) justification right after your answer (even for NA).

**The checklist answers are an integral part of your paper submission.** They are visible to the reviewers, area chairs, senior area chairs, and ethics reviewers. You will be asked to also include it (after eventual revisions) with the final version of your paper, and its final version will be published with the paper.

The reviewers of your paper will be asked to use the checklist as one of the factors in their evaluation. While "[Yes] " is generally preferable to "[No] ", it is perfectly acceptable to answer "[No] " provided a proper justification is given (e.g., "error bars are not reported because it would be too computationally expensive" or "we were unable to find the license for the dataset we used"). In general, answering "[No] " or "[NA] " is not grounds for rejection. While the questions are phrased in a binary way, we acknowledge that the true answer is often more nuanced, so please just use your best judgment and write a justification to elaborate. All supporting evidence can appear either in the main paper or the supplemental material, provided in appendix. If you answer [Yes] to a question, in the justification please point to the section(s) where related material for the question can be found.

IMPORTANT, please:

- **Delete this instruction block, but keep the section heading "NeurIPS Paper Checklist",**
- **Keep the checklist subsection headings, questions/answers and guidelines below.**
- **Do not modify the questions and only use the provided macros for your answers**.

1. **Claims**

   Question: Do the main claims made in the abstract and introduction accurately reflect the paper's contributions and scope?

   Answer: [Yes]

   Justification: The paper's contributions include the KnowMol-100K dataset (Sec.3) and the chemically-informative molecular representation learning methods (Sec.4). We have tried our best to accurately reflect them in the abstract and introduction.

   Guidelines:
   - The answer NA means that the abstract and introduction do not include the claims made in the paper.
   - The abstract and/or introduction should clearly state the claims made, including the contributions made in the paper and important assumptions and limitations. A No or NA answer to this question will not be perceived well by the reviewers.
   - The claims made should match theoretical and experimental results, and reflect how much the results can be expected to generalize to other settings.
   - It is fine to include aspirational goals as motivation as long as it is clear that these goals are not attained by the paper.

2. **Limitations**

   Question: Does the paper discuss the limitations of the work performed by the authors?

   Answer: [Yes]

Justification: We discuss the limitations and the future works in Sec.6.

Guidelines:

- The answer NA means that the paper has no limitation while the answer No means that the paper has limitations, but those are not discussed in the paper.
- The authors are encouraged to create a separate "Limitations" section in their paper.
- The paper should point out any strong assumptions and how robust the results are to violations of these assumptions (e.g., independence assumptions, noiseless settings, model well-specification, asymptotic approximations only holding locally). The authors should reflect on how these assumptions might be violated in practice and what the implications would be.
- The authors should reflect on the scope of the claims made, e.g., if the approach was only tested on a few datasets or with a few runs. In general, empirical results often depend on implicit assumptions, which should be articulated.
- The authors should reflect on the factors that influence the performance of the approach. For example, a facial recognition algorithm may perform poorly when image resolution is low or images are taken in low lighting. Or a speech-to-text system might not be used reliably to provide closed captions for online lectures because it fails to handle technical jargon.
- The authors should discuss the computational efficiency of the proposed algorithms and how they scale with dataset size.
- If applicable, the authors should discuss possible limitations of their approach to address problems of privacy and fairness.
- While the authors might fear that complete honesty about limitations might be used by reviewers as grounds for rejection, a worse outcome might be that reviewers discover limitations that aren't acknowledged in the paper. The authors should use their best judgment and recognize that individual actions in favor of transparency play an important role in developing norms that preserve the integrity of the community. Reviewers will be specifically instructed to not penalize honesty concerning limitations.

3. **Theory assumptions and proofs**

   Question: For each theoretical result, does the paper provide the full set of assumptions and a complete (and correct) proof?

   Answer: [NA]

   Justification: The paper does not include theoretical results.

   Guidelines:

   - The answer NA means that the paper does not include theoretical results.
   - All the theorems, formulas, and proofs in the paper should be numbered and cross-referenced.
   - All assumptions should be clearly stated or referenced in the statement of any theorems.
   - The proofs can either appear in the main paper or the supplemental material, but if they appear in the supplemental material, the authors are encouraged to provide a short proof sketch to provide intuition.
   - Inversely, any informal proof provided in the core of the paper should be complemented by formal proofs provided in appendix or supplemental material.
   - Theorems and Lemmas that the proof relies upon should be properly referenced.

4. **Experimental result reproducibility**

   Question: Does the paper fully disclose all the information needed to reproduce the main experimental results of the paper to the extent that it affects the main claims and/or conclusions of the paper (regardless of whether the code and data are provided or not)?

   Answer: [Yes]

   Justification: We provide details of the model architecture in Sec.4, details of data construction in SecC, and details of training settings in Sec.D.

   Guidelines:

- The answer NA means that the paper does not include experiments.
- If the paper includes experiments, a No answer to this question will not be perceived well by the reviewers: Making the paper reproducible is important, regardless of whether the code and data are provided or not.
- If the contribution is a dataset and/or model, the authors should describe the steps taken to make their results reproducible or verifiable.
- Depending on the contribution, reproducibility can be accomplished in various ways. For example, if the contribution is a novel architecture, describing the architecture fully might suffice, or if the contribution is a specific model and empirical evaluation, it may be necessary to either make it possible for others to replicate the model with the same dataset, or provide access to the model. In general. releasing code and data is often one good way to accomplish this, but reproducibility can also be provided via detailed instructions for how to replicate the results, access to a hosted model (e.g., in the case of a large language model), releasing of a model checkpoint, or other means that are appropriate to the research performed.
- While NeurIPS does not require releasing code, the conference does require all submissions to provide some reasonable avenue for reproducibility, which may depend on the nature of the contribution. For example
  (a) If the contribution is primarily a new algorithm, the paper should make it clear how to reproduce that algorithm.
  (b) If the contribution is primarily a new model architecture, the paper should describe the architecture clearly and fully.
  (c) If the contribution is a new model (e.g., a large language model), then there should either be a way to access this model for reproducing the results or a way to reproduce the model (e.g., with an open-source dataset or instructions for how to construct the dataset).
  (d) We recognize that reproducibility may be tricky in some cases, in which case authors are welcome to describe the particular way they provide for reproducibility. In the case of closed-source models, it may be that access to the model is limited in some way (e.g., to registered users), but it should be possible for other researchers to have some path to reproducing or verifying the results.

5. **Open access to data and code**

   Question: Does the paper provide open access to the data and code, with sufficient instructions to faithfully reproduce the main experimental results, as described in supplemental material?

   Answer: [Yes]

   Justification: We have open-sourced the datasets on HuggingFace and the codes on Github.

   Guidelines:
   - The answer NA means that paper does not include experiments requiring code.
   - Please see the NeurIPS code and data submission guidelines (`https://nips.cc/public/guides/CodeSubmissionPolicy`) for more details.
   - While we encourage the release of code and data, we understand that this might not be possible, so "No" is an acceptable answer. Papers cannot be rejected simply for not including code, unless this is central to the contribution (e.g., for a new open-source benchmark).
   - The instructions should contain the exact command and environment needed to run to reproduce the results. See the NeurIPS code and data submission guidelines (`https://nips.cc/public/guides/CodeSubmissionPolicy`) for more details.
   - The authors should provide instructions on data access and preparation, including how to access the raw data, preprocessed data, intermediate data, and generated data, etc.
   - The authors should provide scripts to reproduce all experimental results for the new proposed method and baselines. If only a subset of experiments are reproducible, they should state which ones are omitted from the script and why.
   - At submission time, to preserve anonymity, the authors should release anonymized versions (if applicable).

- Providing as much information as possible in supplemental material (appended to the paper) is recommended, but including URLs to data and code is permitted.

6. **Experimental setting/details**

   Question: Does the paper specify all the training and test details (e.g., data splits, hyper-parameters, how they were chosen, type of optimizer, etc.) necessary to understand the results?

   Answer: [Yes]

   Justification: We have specified the experiment details in Sec.5.1 and Sec.D.

   Guidelines:

   - The answer NA means that the paper does not include experiments.
   - The experimental setting should be presented in the core of the paper to a level of detail that is necessary to appreciate the results and make sense of them.
   - The full details can be provided either with the code, in appendix, or as supplemental material.

7. **Experiment statistical significance**

   Question: Does the paper report error bars suitably and correctly defined or other appropriate information about the statistical significance of the experiments?

   Answer: [No]

   Justification: Because of the expensive computation cost of LLMs, we adhered to the common practice in the community and did not report the error bars.

   Guidelines:

   - The answer NA means that the paper does not include experiments.
   - The authors should answer "Yes" if the results are accompanied by error bars, confidence intervals, or statistical significance tests, at least for the experiments that support the main claims of the paper.
   - The factors of variability that the error bars are capturing should be clearly stated (for example, train/test split, initialization, random drawing of some parameter, or overall run with given experimental conditions).
   - The method for calculating the error bars should be explained (closed form formula, call to a library function, bootstrap, etc.)
   - The assumptions made should be given (e.g., Normally distributed errors).
   - It should be clear whether the error bar is the standard deviation or the standard error of the mean.
   - It is OK to report 1-sigma error bars, but one should state it. The authors should preferably report a 2-sigma error bar than state that they have a 96% CI, if the hypothesis of Normality of errors is not verified.
   - For asymmetric distributions, the authors should be careful not to show in tables or figures symmetric error bars that would yield results that are out of range (e.g. negative error rates).
   - If error bars are reported in tables or plots, The authors should explain in the text how they were calculated and reference the corresponding figures or tables in the text.

8. **Experiments compute resources**

   Question: For each experiment, does the paper provide sufficient information on the computer resources (type of compute workers, memory, time of execution) needed to reproduce the experiments?

   Answer: [Yes]

   Justification: We provide the computer resources in Sec.D.

   Guidelines:

   - The answer NA means that the paper does not include experiments.
   - The paper should indicate the type of compute workers CPU or GPU, internal cluster, or cloud provider, including relevant memory and storage.

- The paper should provide the amount of compute required for each of the individual experimental runs as well as estimate the total compute.
- The paper should disclose whether the full research project required more compute than the experiments reported in the paper (e.g., preliminary or failed experiments that didn't make it into the paper).

9. **Code of ethics**

Question: Does the research conducted in the paper conform, in every respect, with the NeurIPS Code of Ethics https://neurips.cc/public/EthicsGuidelines?

Answer: [Yes]

Justification: We follow the NeurIPS Code of Ethics.

Guidelines:

- The answer NA means that the authors have not reviewed the NeurIPS Code of Ethics.
- If the authors answer No, they should explain the special circumstances that require a deviation from the Code of Ethics.
- The authors should make sure to preserve anonymity (e.g., if there is a special consideration due to laws or regulations in their jurisdiction).

10. **Broader impacts**

Question: Does the paper discuss both potential positive societal impacts and negative societal impacts of the work performed?

Answer: [Yes]

Justification: We discuss the potential societal impacts in Sec.6.

Guidelines:

- The answer NA means that there is no societal impact of the work performed.
- If the authors answer NA or No, they should explain why their work has no societal impact or why the paper does not address societal impact.
- Examples of negative societal impacts include potential malicious or unintended uses (e.g., disinformation, generating fake profiles, surveillance), fairness considerations (e.g., deployment of technologies that could make decisions that unfairly impact specific groups), privacy considerations, and security considerations.
- The conference expects that many papers will be foundational research and not tied to particular applications, let alone deployments. However, if there is a direct path to any negative applications, the authors should point it out. For example, it is legitimate to point out that an improvement in the quality of generative models could be used to generate deepfakes for disinformation. On the other hand, it is not needed to point out that a generic algorithm for optimizing neural networks could enable people to train models that generate Deepfakes faster.
- The authors should consider possible harms that could arise when the technology is being used as intended and functioning correctly, harms that could arise when the technology is being used as intended but gives incorrect results, and harms following from (intentional or unintentional) misuse of the technology.
- If there are negative societal impacts, the authors could also discuss possible mitigation strategies (e.g., gated release of models, providing defenses in addition to attacks, mechanisms for monitoring misuse, mechanisms to monitor how a system learns from feedback over time, improving the efficiency and accessibility of ML).

11. **Safeguards**

Question: Does the paper describe safeguards that have been put in place for responsible release of data or models that have a high risk for misuse (e.g., pretrained language models, image generators, or scraped datasets)?

Answer: [No]

Justification: We follow the PubChem and vicuna licenses and hope that the users to follow them too. But due to the open-source property, it is challenging for us to provide comprehensive safeguards.

Guidelines:

- The answer NA means that the paper poses no such risks.
- Released models that have a high risk for misuse or dual-use should be released with necessary safeguards to allow for controlled use of the model, for example by requiring that users adhere to usage guidelines or restrictions to access the model or implementing safety filters.
- Datasets that have been scraped from the Internet could pose safety risks. The authors should describe how they avoided releasing unsafe images.
- We recognize that providing effective safeguards is challenging, and many papers do not require this, but we encourage authors to take this into account and make a best faith effort.

12. **Licenses for existing assets**

Question: Are the creators or original owners of assets (e.g., code, data, models), used in the paper, properly credited and are the license and terms of use explicitly mentioned and properly respected?

Answer: [Yes]

Justification: We have cited, acknowledged, and followed the license of the creators or original owners of assets used in the paper.

Guidelines:

- The answer NA means that the paper does not use existing assets.
- The authors should cite the original paper that produced the code package or dataset.
- The authors should state which version of the asset is used and, if possible, include a URL.
- The name of the license (e.g., CC-BY 4.0) should be included for each asset.
- For scraped data from a particular source (e.g., website), the copyright and terms of service of that source should be provided.
- If assets are released, the license, copyright information, and terms of use in the package should be provided. For popular datasets, `paperswithcode.com/datasets` has curated licenses for some datasets. Their licensing guide can help determine the license of a dataset.
- For existing datasets that are re-packaged, both the original license and the license of the derived asset (if it has changed) should be provided.
- If this information is not available online, the authors are encouraged to reach out to the asset's creators.

13. **New assets**

Question: Are new assets introduced in the paper well documented and is the documentation provided alongside the assets?

Answer: [Yes]

Justification: We have documented the new assets in this paper and provided alongside the assets on the huggingface.

Guidelines:

- The answer NA means that the paper does not release new assets.
- Researchers should communicate the details of the dataset/code/model as part of their submissions via structured templates. This includes details about training, license, limitations, etc.
- The paper should discuss whether and how consent was obtained from people whose asset is used.
- At submission time, remember to anonymize your assets (if applicable). You can either create an anonymized URL or include an anonymized zip file.

14. **Crowdsourcing and research with human subjects**

Question: For crowdsourcing experiments and research with human subjects, does the paper include the full text of instructions given to participants and screenshots, if applicable, as well as details about compensation (if any)?

Answer: [NA]

Justification: We do not include crowdsourcing experiments or research with human subjects.

Guidelines:

- The answer NA means that the paper does not involve crowdsourcing nor research with human subjects.
- Including this information in the supplemental material is fine, but if the main contribution of the paper involves human subjects, then as much detail as possible should be included in the main paper.
- According to the NeurIPS Code of Ethics, workers involved in data collection, curation, or other labor should be paid at least the minimum wage in the country of the data collector.

15. **Institutional review board (IRB) approvals or equivalent for research with human subjects**

Question: Does the paper describe potential risks incurred by study participants, whether such risks were disclosed to the subjects, and whether Institutional Review Board (IRB) approvals (or an equivalent approval/review based on the requirements of your country or institution) were obtained?

Answer: [NA]

Justification: We do not include crowdsourcing experiments or research with human subjects.

Guidelines:

- The answer NA means that the paper does not involve crowdsourcing nor research with human subjects.
- Depending on the country in which research is conducted, IRB approval (or equivalent) may be required for any human subjects research. If you obtained IRB approval, you should clearly state this in the paper.
- We recognize that the procedures for this may vary significantly between institutions and locations, and we expect authors to adhere to the NeurIPS Code of Ethics and the guidelines for their institution.
- For initial submissions, do not include any information that would break anonymity (if applicable), such as the institution conducting the review.

16. **Declaration of LLM usage**

Question: Does the paper describe the usage of LLMs if it is an important, original, or non-standard component of the core methods in this research? Note that if the LLM is used only for writing, editing, or formatting purposes and does not impact the core methodology, scientific rigorousness, or originality of the research, declaration is not required.

Answer: [Yes]

Justification: We use LLM as an important component, and we describe the usage of the LLM in Sec.3 and Sec.A.

Guidelines:

- The answer NA means that the core method development in this research does not involve LLMs as any important, original, or non-standard components.
- Please refer to our LLM policy (`https://neurips.cc/Conferences/2025/LLM`) for what should or should not be described.

**Structural Construction Level Prompt:**

Given the molecule's SMILES: {smiles}, IUPAC name: {IUPACName}, a picture of the molecular structure, and its functional groups: {fg_names}. Please analyze the composition of the molecule based on these information, without missing any substructures, including the main chain, side chain, ring, and other substructures or functional groups, and then analyze the connections between every substructures or functional groups step by step.\nRequirements: Pay attention to utilize the given molecular picture, distinguish the names of different substructures when analyzing substructures, do not make factual errors when analyzing the connection between functional groups, and do not produce structural analysis that is inconsistent with the structure shown in the picture and SMILES formula, and do not make quantitative errors when analyzing the number of atoms in the substructure. Please do not include irrelevant information other than the molecular structure, especially do not include content related to the properties of the molecule. Please output unambiguous analyses in the simplest sentence structure possible, without including complex IUPAC names in sentences. Please answer in one paragraph. Please do not repeat the SMILES and IUPAC name of the molecule in your answer.

**Physicochemical Property Level Prompt:**

Property Analysis Guideline

Polarity: The polarity of a molecule is generally affected by its structure or substructure through the arrangement of atoms and the shape of the molecule. Even if a molecule has polar bonds, its overall polarity depends on whether these bond dipoles cancel out or reinforce each other. Symmetrical structures tend to be nonpolar, while asymmetrical structures with uneven charge distribution lead to polar molecules.

Acidity or Basicity: The acidity or basicity of a molecule is affected by its structure or substructure through the presence of electron-donating or electron-withdrawing groups. Electron-withdrawing groups stabilize negative charges, increasing acidity. Electron-donating groups decrease acidity and increase basicity by stabilizing positive charges. Additionally, resonance, inductive effects, and the hybridization of atoms involved in the acidic or basic site can also influence acidity and basicity.

Solubility: The solubility of a molecule is generally affected by its structure or substructure through the presence of polar or nonpolar groups. Polar groups (e.g., hydroxyl, amine) enhance solubility in polar solvents like water, while nonpolar groups (e.g., alkyl, aromatic rings) increase solubility in nonpolar solvents. The size and branching of the molecule also play roles—smaller and more branched molecules tend to be more soluble due to better interactions with the solvent.

Reactivity: The reactivity of a molecule is generally affected by its structure or substructure through the presence of functional groups, electron density, and strain. Reactive functional groups (e.g., carbonyl, hydroxyl) dictate the types of chemical reactions a molecule can undergo. Electron-withdrawing or electron-donating groups influence electron density at reactive sites, making them more or less reactive. Additionally, structural strain (e.g., in rings) can increase reactivity by making bonds easier to break.

Stereochemistry: The stereochemistry of a molecule is affected by its structure or substructure through the presence of chiral centers, double bonds, and ring structures. Chiral centers result in different enantiomers, which are non-superimposable mirror images. Double bonds can lead to cis/trans isomerism based on the spatial arrangement around the bond. Ring structures can create different conformations and affect the overall 3D shape of the molecule.

Electrophilicity: The electrophilicity of a molecule is generally affected by its structure or substructure through the presence of electron-withdrawing groups and the overall electron density around the electrophilic center. Electron-withdrawing groups (e.g., carbonyl, nitro) increase electrophilicity by making the electrophilic center more positively charged or electron-deficient. The nature of the electrophilic site, such as a partially positive carbon in a carbonyl group, also influences reactivity.

Given the molecule's SMILES: {smiles}, IUPAC name: {IUPACName}, a picture of the molecular structure, functional group: {fg_names}, and its structural construction: {mol_messages['construction']}. Based on these information, please consider the effects of functional groups and molecular structure on the properties to analyze the physical and chemical properties of the molecule, including: Polarity, Acidity or Basicity, Solubility, Reactivity, Stereochemistry, Electrophilicity. \nRequirements: Do not make factual errors, do not confuse different causal relationships, and output unambiguous analysis in the simplest possible sentence structure. Please answer in one paragraph. Please do not repeat the molecule's IUPAC name in your answer.

Figure 5: Prompt for generating structural construction and physicochemical property descriptions to construct KnowMol-100K.

## A  GPT-generated annotations

To guarantee the correctness and mitigate the errors in the GPT-generated annotations for constructing KnowMol-100K, we designed a comprehensive annotation prompt. We visualize the prompt in Fig.5.

In the prompt, each set of brackets is filled with corresponding information. We provided multi-source data with diverse descriptive perspectives to maximize the reference information available to GPT-4o for generating annotations. Additionally, we designed detailed instructional cues to ensure the accuracy and efficiency of GPT-4o's generation.

# B  Dataset Quality Inspection

To evaluate the quality of the generated chemical molecular descriptions in KnowMol-100K, we randomly selected a subset of size 30 for in-depth assessment. In detail, we carefully designed an evaluation criterion to ensure scientific rigor and invited three chemistry expert volunteers to validate the dataset according to the criteria.

Given the molecule's SMILES, IUPAC name, and picture of the molecular structure. GPT4 is asked to analyze the composition of the molecule based on these information, without missing any substructures, including the main chain, side chain, ring, and other substructures or functional groups, and analyze the connections between every substructures or functional groups step by step.

Your task is to evaluate the molecular structure description text generated by GPT4. The evaluation dimensions for the structural description are as follows.

*****Evaluation Dimensions for the Structural Description*****
1. Factual Accuracy
Recognition of atoms and functional groups: Does the description correctly identify all atoms, functional groups, and ring systems in the molecule without omissions or additions?
Accuracy of connectivity: Are the connections between each group or substructure in the molecule accurately described?

2. Completeness
Coverage of substructures: Does the description cover all required and important substructures, such as the main chain, side chains, ring systems, and functional groups? Are certain rings/bridged rings or key functional groups (e.g., ester, amine, ether) that clearly exist in the molecule omitted?
Completeness of elements: Does the description mention all substructures (alkenyl chains, ring systems, ester groups, aromatic rings, amine groups, etc.) and their interconnections?

3. Clarity and Conciseness
Concise and unambiguous: Does the description follow the requirement to be written in a single paragraph with straightforward sentence structures, free of excessive detail or overly complex technical terms?
No unnecessary repetition: Does the text avoid repetition or contain redundant information?

*****Scoring Reference*****
You should assign a score of poor/acceptable/excellent to each dimension. For example:
 For example:

Structural Description Dimensions
Factual Accuracy: excellent
Completeness: acceptable
Clarity and Conciseness: acceptable

Figure 6: structural construction evaluation criteria for chemistry experts. The criteria include three aspects: Factual Accuracy, Completeness, Clarity and Conciseness.

The evaluation criteria are shown in Fig 6 and Fig 7. The evaluation criteria for structural construction include three aspects: Factual Accuracy, Completeness, Clarity and Conciseness. The criteria for physicochemical property description include four aspects: Factual Accuracy, Completeness, Consistency, Clarity and Conciseness. To better demonstrate the quality of KnowMol-100K, we assign a score for each level, 0 for poor, 1 for fair, 2 for acceptable, and 3 for excellent. The average score of each aspect is shown in Table 7.

Table 7: Average score of each aspect. We assign a score for each level, 1 for poor, 2 for acceptable, and 3 for excellent.

| Structural Description | | | Property Description | | | | Overall |
|---|---|---|---|---|---|---|---|
| Factual Accuracy | Completeness | Clarity and Conciseness | Factual Accuracy | Completeness | Consistency | Clarity and Conciseness | |
| 2.10 | 2.33 | 2.38 | 2.48 | 2.24 | 2.62 | 2.43 | 2.43 |

The evaluation of KnowMol-100K demonstrates its strong reliability and quality, with an overall average score of 2.43 out of 3, reflecting its effectiveness in generating chemical molecular descriptions. The dataset excels in ensuring consistency, particularly in property descriptions, which achieved the highest score of 2.62, highlighting its robustness in maintaining uniformity and logical structure. Additionally, the clarity and conciseness of descriptions received commendable scores (2.38 for

Given the molecule's SMILES, IUPAC name, a picture of the molecular structure, and its structural construction generated by GPT4 previously. Based on these information, GPT4 is asked to consider the effects of functional groups and molecular structure on the properties to analyze the physical and chemical properties of the molecule, including: Polarity, Acidity or Basicity, Solubility, Reactivity, Stereochemistry, Electrophilicity.

Your task is to evaluate the molecular property description text generated by GPT4. The evaluation dimensions for the property description are as follows.

Evaluation Dimensions for the Property Description
1. Factual Accuracy
Analysis of functional groups' influence on properties: Does the description correctly explain how functional groups such as ester, amine, and aromatic rings affect molecular polarity, acidity/basicity, solubility, etc.?
Errors in qualitative features: For example, does the text describe a portion of the molecule as acidic when it is obviously not, or incorrectly emphasize a non-existent hydrogen bonding?

2. Logic and Causality
Causal relationships: Does the text correctly identify which functional groups lead to specific property changes, or explicitly clarify the causes of certain properties? For instance, claiming "the molecule has an amine group → the molecule shows acidity" is an obvious causal error and a serious mistake.
Reactivity and structural interpretation: Does the text provide a reasonable structural explanation for which parts of the molecule might be more reactive toward nucleophilic/electrophilic/radical reactions?

3. Completeness
Coverage of the specified properties: Does the description address the requested properties such as polarity, acidity/basicity, solubility, reactivity, stereochemistry, and electrophilicity?
No critical omissions: For instance, failing to mention notable steric hindrance or ignoring multiple chiral centers with potential mixtures of stereoisomers.

4. Consistency
Content alignment: Do the properties described correspond to the structural analysis, with no conflict between the two? For example, if the structural description mentions a primary amine, does the property description mistakenly referring to it as a secondary amine?

5. Clarity and Conciseness
Readable and straightforward: Does the text use simple sentences to describe properties, avoiding excessive complexity or unnecessary technical jargon?
Avoidance of redundancy: Does it refrain from unrelated structural details, and avoid extending into irrelevant topics?

Scoring Reference
You should assign a score of poor/acceptable/excellent to each dimension. For example:

Property Description Dimensions
Factual Accuracy: excellent
Logic and Causality: acceptable
Completeness: poor
Consistency: acceptable
Clarity and Conciseness: excellent

Figure 7: physicochemical property description evaluation criteria for chemistry experts. The criteria include four aspects: Factual Accuracy, Completeness, Consistency, Clarity and Conciseness.

structural descriptions and 2.43 for property descriptions), showcasing its ability to present complex information in an accessible manner. These results affirm the scientific rigor of KnowMol-100K and its potential as a valuable resource for downstream applications in the field of chemistry.

## C  Details of Datasets

We provide a summary of the datasets involved in our paper, including our constructed dataset KnowMol-100K and the downstream task datasets.

### C.1  KnowMol-100K

In KnowMol-100K, there are 1,000,000 molecules selected from the PubChem database annotated with four level fundamental chemical understanding factors. Table 8 shows the statistics of these factors.

Table 8: Average word of fundamental factors for molecule understanding in the description from KnowMol-100K.

| dataset | atoms | Functional groups | molecular structure | Physicochemical property | | | | | | | full description |
|---|---|---|---|---|---|---|---|---|---|---|---|
| | | | | Polarity | Acidity/Basicity | Solubility | Reactivity | Stereochemistry | Electrophilicity | Sum | |
| KnowMol-100K | 7.198 | 14.339 | 161.780 | 58.840 | 25.282 | 30.435 | 40.414 | 21.426 | 32.141 | 208.538 | 391.855 |

As illustrated in Table 1 and Table 8, the KnowMol-100K dataset demonstrates a substantially broader and deeper coverage of molecular characteristics compared to the PubChem dataset. Notably, the average word count for key molecular factors in KnowMol-100K is significantly higher across all categories. For instance, molecular structure, a critical factor for molecular understanding, receives an extensive average coverage of 161.780 words in KnowMol-100K, whereas it is limited to merely 2.406 words in PubChem's sample. This disparity underscores the deliberate focus of KnowMol-100K on providing detailed and nuanced descriptions. This level of detail is evident in the total average word count for KnowMol-100K (391.855), which greatly surpassed the 19.338 words seen in PubChem's sample.

KnowMol-100K also excels in representing factors that are severely underrepresented in PubChem. Properties such as polarity and electrophilicity, which are nearly absent in PubChem captions, achieve robust coverage in KnowMol-100K with average word counts of 58.840 and 32.141, respectively. This comprehensive representation ensures that KnowMol-100K addresses critical aspects of molecular characterization that are overlooked in PubChem, thereby offering a more balanced and holistic dataset.

## C.2 Downstream Datasets

Table 9: Examples of instruction following data on each downstream task.

| task | qusetion | answer |
|---|---|---|
| molecule captioning | Could you provide a description of this molecule? The compound SELFIES sequence is: [SELFIES] | The molecule is an indole phytoalexin that is indole substituted at position 3 by ...... |
| molecule property prediction (classification) | BACE1 plays a significant role in the development of Alzheimer's disease and the creation of myelin sheaths as an essential aspartic-acid protease. Is it possible for this molecule to attach to BACE1? The compound SELFIES sequence is: [SELFIES] | Yes |
| molecule property prediction (regression) | Please provide the energy separation between the highest occupied and lowest unoccupied molecular orbitals (HOMO-LUMO gap) of this molecule. The compound SELFIES sequence is: [SELFIES] | 0.1913 |
| caption-guided molecule generation | Create a molecule with the structure as the one described. The molecule's description is: The molecule is a natural product found in Picea abies, Citrus unshiu, and other organisms with data available. | [SELFIES] |
| reagent prediction | Based on the given chemical reaction, can you propose some likely reagents that might have been utilized? $\langle reactantA \rangle.\langle reactantB \rangle... \gg \langle productA \rangle.\langle productB \rangle...$ | [SELFIES] |
| forward reaction prediction | Please suggest a potential product based on the given reactants and reagents. $\langle reactantA \rangle.\langle reactantB \rangle...\langle reagentA \rangle.\langle reagentB \rangle...$ | [SELFIES] |
| retrosynthesis prediction | Provided the product below, propose some possible reactants that could have been used in the reaction. $\langle productA \rangle.\langle productB \rangle...$ | [SELFIES] |

This section provides detailed information about the downstream datasets used to evaluate the ability of KnowMol. The datasets include molecule captioning datasets, molecule property prediction datasets, and molecule generation datasets. We provide some examples of the instruction following data on each downstream task in Table 9.

**Details of molecule captioning datasets.** The molecule captioning datasets mainly focus on generating the corresponding description of a given molecule. In this task, we use a widely used dataset ChEBI-20 [9]. This dataset contains 33,010 molecule-description pairs longer than 20 words selected from the PubChem database. The molecule-description pairs in ChEBI-20 are separated into train, validation, and test splits in 80%, 10% and 10%. Based on the original dataset and the splitting, we transform the molecule-description pairs into instruction following form.

**Details of molecule property prediction datasets.** The molecule property prediction tasks are designed to predict the given molecule's specific chemical or physical properties. For this task, we consider both binary classification tasks and regression tasks on molecule property prediction.

For binary classification tasks, we use five datasets derived from MoleculeNet [50], BACE, BBBP, HIV, MUV and Tox21. The BACE dataset aims to predict the inhibitors of the BACE-1 enzyme. The BBBP dataset aims to predict whether the given molecule is able to penetrate the blood-brain barrier. The HIV dataset aims to predict whether the given molecule can impede the replication of the HIV virus. The MUV dataset is selected from PubChem BioAssay and contains 17 tasks for around 90,000 compounds which aims to the validation of virtual screening techniques. The Tox21 dataset contains toxicity measurements for 8k compounds on 12 different targets, and aims to measure the toxicity of compounds. We use the scaffold splitting to split the dataset and transform the dataset into instruction following data using the prompt from GIMLET [54].

For regression tasks, we consider the QM9 dataset. This dataset aims to predict the quantum mechanics properties, of the molecules. The quantum mechanics properties include: (1) Highest occupied molecular orbital (HOMO) energy; (2) Lowest occupied molecular orbital (LUMO) energy; (3) and HUMO-LUMO gap energy. We adopt the process dataset of Mol-Instructions [15].

**Details of molecule generation datasets.** For the molecule generation datasets, we consider both caption-guided molecule generation and chemical reaction prediction tasks. The chemical reaction prediction tasks involve three subtasks: reagent prediction, forward reaction prediction, and retrosynthesis prediction. All of the four tasks aim to predict the corresponding molecules according to the given condition. We adopt the four datasets from Mol-Instructions [15].

## D   Details of Training

**Architecture.** We use a pretrained GNN by MoleculeSTM [28] as the basic molecule graph encoder. The molecule graph encoder is a 5-layer GIN with a hidden dimension of 300. The multi-level feature projection includes three single-layer MLPs corresponding to three hierarchies, used to connect the molecule and text modality. The LLM is the open-source vicuna-v-1.3-7B [5]. The overall scale of parameters of KnowMol is around 6.9B. The input of the model includes both 1D SELFIES string and 2D molecule graph.

**Training settings.** To enable fair comparisons, we adopted the training parameter settings consistent with the baselines [2, 3, 16]. For the LoRA adapters [18] used in the two stage tuning, we use a LoRA rank of 64, a scaling value $\alpha$ of 256, and dropout 0.1 for all of the training stage and training tasks. All experiments are run with 8×RTX A40 (48GB) GPUs.

In pretraining stage, we train the model using two tasks constructed on KnowMol-100K. we conduct the training for 5 epochs, with batch size 64, learning rate 8e-5, weight decay 0.05 and warmup ratio 3%.

For molecule captioning dataset, we conduct the training for 50 epochs, with batch size 64, learning rate 8e-5, weight decay 0.05 and warmup ratio 3%.

For classification molecule property prediction datasets, we conduct the training for 10 epochs, with batch size 64, learning rate 8e-5, weight decay 0 and warmup ratio 3%.

For regression molecule property prediction datasets, we conduct the training for 15 epochs, with batch size 128, learning rate 8e-5, weight decay 0.05 and warmup ratio 3%.

For four molecule generation tasks: caption-guided molecule generation datasets, reagent prediction datasets, forward reaction prediction datasets, and retrosynthesis prediction datasets. We conduct the training for 15 epochs, with batch size 64, learning rate 8e-5, weight decay 0 and warmup ratio 3%.

## E   Pre-training Cost analysis

We provide a detailed computation cost comparison between our KnowMol and the second-best model UniMoT [16] in Table 10. Under identical downstream task settings, our analysis focuses on pre-training costs, the primary difference between methods. KnowMol requires only 5 epochs on 200K samples built from KnowMol-100K and achieves SOTA performance, while UniMoT needs 3 training stages with more epochs and larger datasets.

Table 10: Computation cost comparison of the pretrain stage(s) between KnowMol and UniMoT.

| model | pre-training stage | training data | data size | training epoch |
|---|---|---|---|---|
| UniMoT | Causal Q-Former Pretraining | PubChem pretrain subset | 301658 | 50 |
| | Molecule Tokenizer Pretraining | PubChem pretrain subset, CheBI-20 train subset | 328065 | 50 |
| | Unified Molecule-Text Pretraining | PubChem pretrain subset, CheBI-20 train subset | 328065 | 10 |
| KnowMol | pertaining on KnowMol-100K | KnowMol-100K | 200000 | 5 |

# F   Effect of LoRA

We evaluate LoRA's impact through an ablation study on the molecular property prediction tasks (regression) in Table 11. Given the prohibitive computational cost of full LLM fine-tuning, we focus on optimizing LoRA hyperparameters. The results demonstrate that LoRA provides substantial improvements, since higher LoRA capacity enables the model to extract more chemical information from KnowMol-100K, thereby enhancing prediction accuracy.

Table 11: Ablation of LoRA on molecular property prediction tasks (regression).

| Rank | scaling $\alpha$ | HOMO ↓ | LUMO ↓ | $\Delta\epsilon$ ↓ | Avg ↓ |
|---|---|---|---|---|---|
| 8 | 32 | 0.0051 | 0.0051 | 0.0064 | 0.0055 |
| 64 | 16 | 0.0039 | 0.0039 | 0.0051 | 0.0043 |
| 64 | 128 | 0.0034 | 0.0037 | 0.0042 | 0.0038 |
| 64 | 256 | 0.0028 | 0.0029 | 0.0034 | 0.0030 |

# G   Qualitative Analysis

We provide three additional qualitative analysis in Fig 8, 9 and 10. Obviously, KnowMol demonstrates significant advantages over InstructMol in the ability to analyze molecules' basic information.

In terms of atomic composition, KnowMol consistently provides accurate molecular formulas, even for complex structures, while InstructMol struggles in this. When identifying functional groups, KnowMol shows the advantage of providing accurate and comprehensive identification. In contrast, InstructMol provides limited analysis and broad coverage, which may hinder a thorough understanding of the molecule. In the structural analysis, KnowMol effectively captures complex molecular geometries and stereochemical features, such as chiral centers and bonding configurations, which are often overlooked by InstructMol. Similarly, in property analysis, KnowMol links structural features to chemical behaviors, such as reactivity and solubility, with greater accuracy and depth than InstructMol. This ability to integrate functional and structural data makes KnowMol a superior tool for advanced molecular research.

In summary, KnowMol surpasses InstructMol across all aspects of molecular understanding—atomic composition, functional groups, structure, and properties—by providing precise, detailed, and application-focused insights. These capabilities position KnowMol as a more powerful and reliable model for advanced chemical analysis.

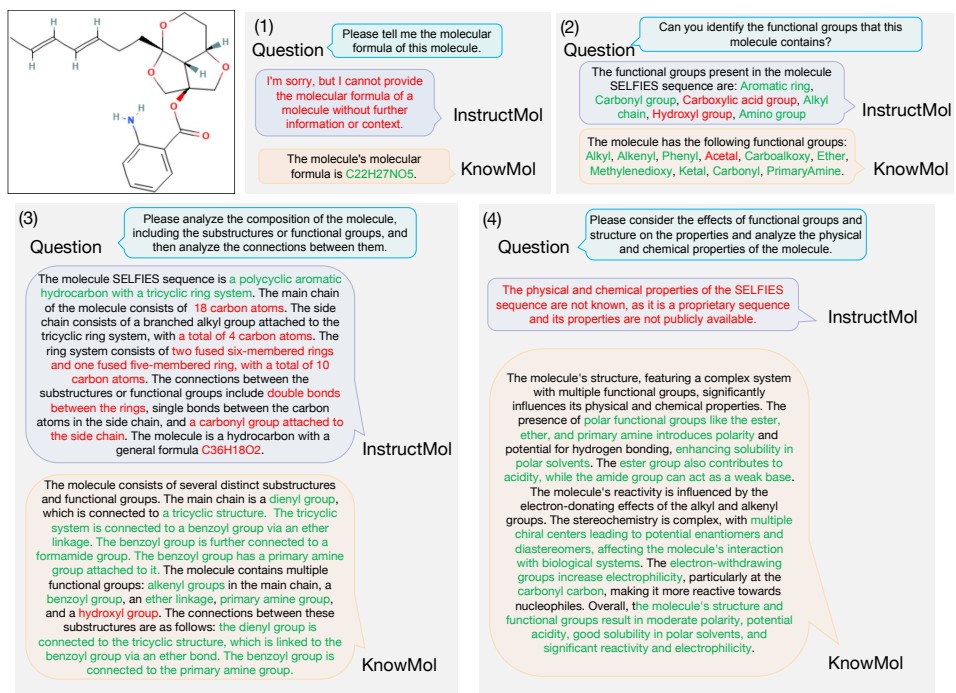

Figure 8: Qualitative Results 1. We mark the wrong/illusion parts with red, the unverifiable parts with orange, and the correct parts with green.

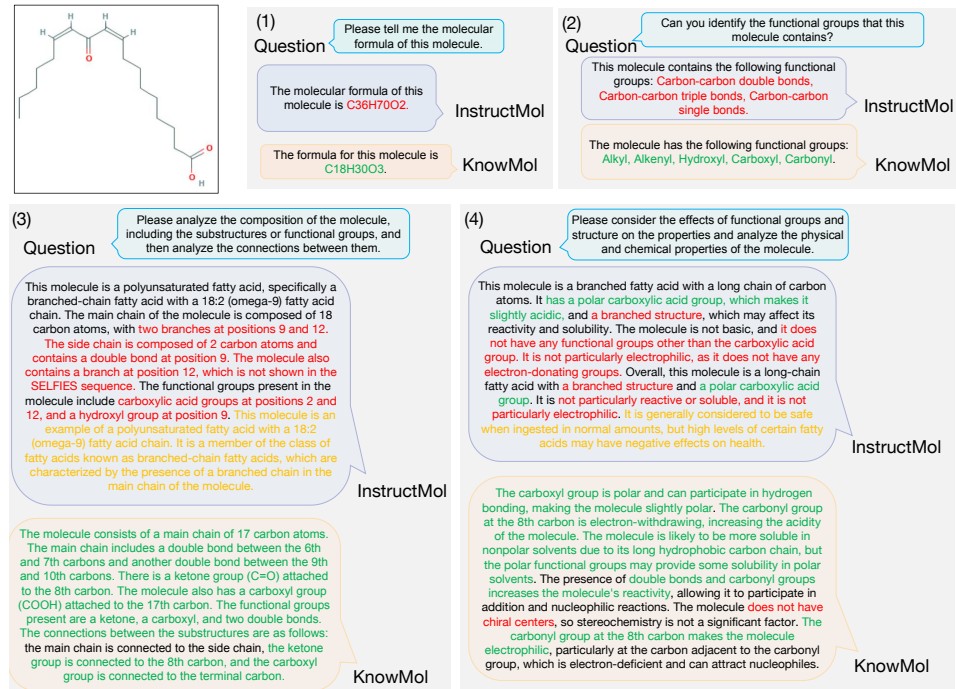

Figure 9: Qualitative Results 2. We mark the wrong/illusion parts with red, the unverifiable parts with orange, and the correct parts with green.

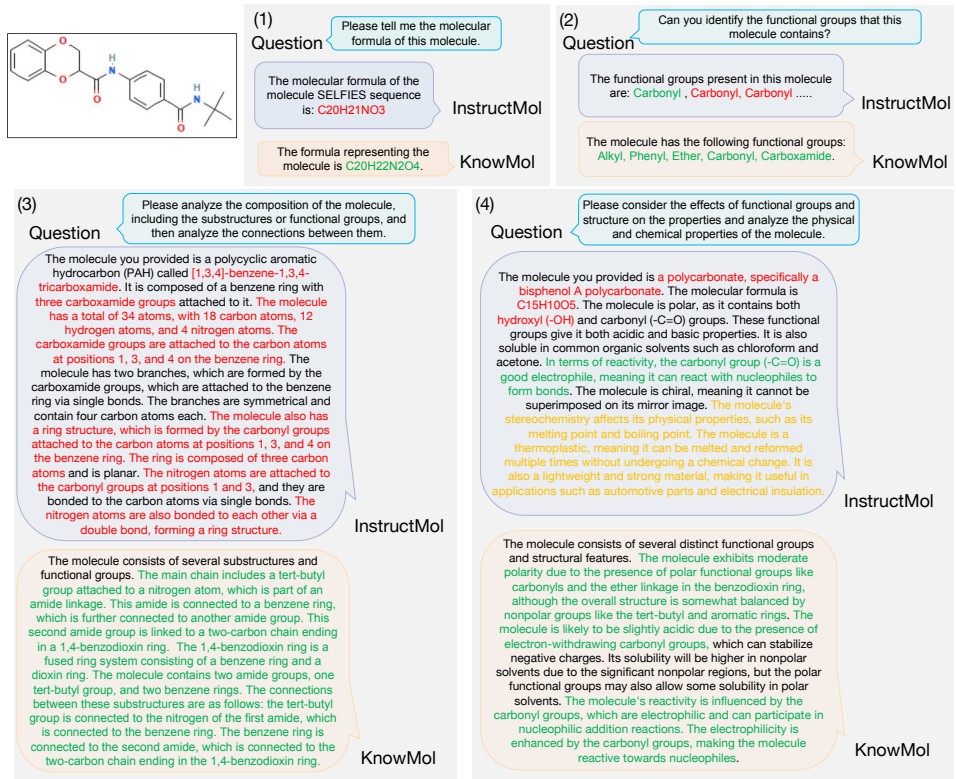

Figure 10: Qualitative Results 3. We mark the wrong/illusion parts with red, the unverifiable parts with orange, and the correct parts with green.

