# OpenReview forum: "KnowMol: Advancing Molecular Large Language Models with Multi-Level Chemical Knowledge"
_NeurIPS.cc/2025/Datasets_and_Benchmarks_Track — NeurIPS 2025 Datasets and Benchmarks Track poster_

### Official Review · Reviewer_Cj73 · 2025-06-24

**Rating:** 4
**Confidence:** 3

**Summary:**

The performance of molecular large models is affected by insufficient text descriptions and suboptimal molecular representation strategies during pre-training. To address the above issues, this paper first constructs a multi-level annotation dataset KnowMol-100K containing 100k molecules to enrich the granularity and coverage of the training data in molecular description, and builds two instruction-following pre-training tasks on this basis. The experimental part shows the leading performance of KnowMol in multiple molecular understanding and generation tasks, significantly exceeding existing Mol-LLMs, Retrieval Based LLMs and LLM-Based Generalist Models, verifying the effectiveness and wide applicability of the proposed method.

**Dataset Code Accessibility:**

Yes

**Dataset Code Comments:**

The dataset is fully accessible and publicly released on Hugging Face, with a clear URL provided in the paper. The authors describe the data collection, annotation, and filtering processes in detail, and the overall pipeline appears reproducible.

**Ethical Comments:**

This dataset is derived entirely from public repositories and does not contain any personally identifiable information or sensitive content. The generated text is based on chemical structure and property annotations and does not involve human subjects, private data, or real-world behavioral information. There are no concerns about informed consent, privacy protection, or legal compliance in the paper. In addition, the authors clearly explain the data source, screening, and construction process, and the dataset has been released for scientific research use in a responsible manner. Therefore, I do not think that additional ethical review is necessary.

**Ethical Considerations:**

No, there are no or only very minor ethics concerns

**Final Justification:**

The author introduces knowmol-100k, which is a smaller dataset than PubChem, but has fine-grained molecular annotation across multiple levels, and provides a strategy for molecular characterization of chemical information. I think the author's work is perfect and contributes to the community.


The author solved most of my puzzles in his reply, especially the discussion about the quality of data sets and the complexity of models. The Ablation Experiment of the author also answered some of my questions about the comparison of baseline models, but the expression is still slightly weak. Therefore, I decided to give rating 4.

**Limitations Weaknesses:**

1.  Most of the annotations at the structural and physicochemical property levels in the dataset are automatically generated by GPT-4o (Section 2.3.2). Although the authors mention expert quality control (Section 2.3.3), the lack of quantitative consistency assessment or error analysis makes me worried about the molecular descriptions in their dataset.

2.  The paper does not currently evaluate the fine-tuning results of other strong baseline models (such as InstructMol and UniMoT) on the KnowMol-100K dataset, so it is difficult to determine whether the performance improvement is due to model improvement or data advantage.

3. The paper does not seem to report the complexity of model training. Since the KnowMol-100K dataset introduces more molecular descriptions, the authors should accurately explain the impact this has on model deployment.

**Strengths Contributions:**

1.  Construct a high-quality molecular multi-layer annotation dataset, KnowMol-100K, covering four levels: atoms, functional groups, structural composition, and physical and chemical properties.

2. In 1D representation, SELFIES is introduced to replace SMILES, and an independent tokenizer is designed; in 2D representation, functional groups are extracted based on BRICS to enhance the chemical information of molecular representation.

3. The proposed model KnowMol has excellent performance and outperforms existing mainstream models in multiple molecular tasks (including molecular description generation, molecular property prediction, reaction prediction and retrosynthesis, etc.) .

---

> ### Author Rebuttal · Authors · 2025-07-30
>
> ####
>
> > Q1: Most of the annotations at the structural and physicochemical property levels in the dataset are automatically generated by GPT-4o (Section 2.3.2). Although the authors mention expert quality control (Section 2.3.3), the lack of quantitative consistency assessment or error analysis makes me worried about the molecular descriptions in their dataset.
>
> R1: We thank the reviewer for raising this important concern regarding the reliability and consistency of automatically generated annotations in our dataset.
>
> We respectfully clarify that a **quantitative consistency assessment** between GPT-4o-generated annotations and chemical expert-level expectations has already been conducted and reported in **Appendix C** of the manuscript.
>
> Specifically, we invited chemistry experts to evaluate a sub-sample of molecular structural and property descriptions. The evaluation was based on clearly defined criteria in Appendix C: *Factual Accuracy, Completeness, Consistency,* *Clarity and Conciseness*. This procedure explicitly assesses the degree of agreement between machine-generated annotations and expert knowledge. The assessment demonstrates a high level of annotation quality, supporting the reliability of the automatically generated annotations.
>
> |                  | Structural Description |                         |                  | Property Description |             |                         | Overall |
> | ---------------- | ---------------------- | ----------------------- | ---------------- | -------------------- | ----------- | ----------------------- | ------- |
> | Factual Accuracy | Completeness           | Clarity and Conciseness | Factual Accuracy | Completeness         | Consistency | Clarity and Conciseness |         |
> | 2.13             | 2.40                   | 2.53                    | 2.60             | 2.20                 | 2.66        | 2.26                    | 2.40    |
>
> To further strengthen our findings,  **we conducted an additional independent validation on 30 new randomly sampled entries from the dataset.** All samples were evaluated by the same domain expert using identical criteria. The results reaffirmed the high factual correctness, completeness, and descriptive clarity of the generated annotations.
>
> In addition to the quantitative consistency assessment, we conducted a **comprehensive qualitative error analysis**, as presented in **Appendix H**, to evaluate the molecular understanding capabilities of the KnowMol model trained on our dataset.
>
> In this analysis, we systematically examined the model’s generated descriptions across key dimensions of molecular understanding: atomic composition, functional groups, structural Construction, and physicochemical properties. Representative cases were selected and manually analyzed to identify typical strengths and failure modes.
>
> The results show that KnowMol demonstrates **high accuracy and consistency** in capturing detailed molecular information, with only **minor and infrequent errors** observed. The model can correctly identify and describe structural constructions and physicochemical attributes in most cases.
>
> These findings indicate that the training data—particularly the GPT-4o-generated annotations—are **effective in equipping the model with robust, multi-level molecular understanding**, thereby validating the quality and utility of the dataset from both qualitative and practical performance perspectives.
>
>
>
> > Q2: The paper does not currently evaluate the fine-tuning results of other strong baseline models (such as InstructMol and UniMoT) on the KnowMol-100K dataset, so it is difficult to determine whether the performance improvement is due to model improvement or data advantage.
>
> R2: We appreciate the reviewer’s insightful comment. To clarify, **InstructMol is indeed included as the baseline of our study (line 177)**. Specifically, the sixth row of Table 6 corresponds to the InstructMol model fine-tuned on KnowMol-100K, but without the enhanced representation strategies proposed in our work. To address the reviewer's concern more clearly, we summarize the relevant settings in Table A below to highlight the respective contributions of the dataset and model improvements.
>
> **Table A: Summary of Table 6 Rows and Their Comparison Purpose**
>
> | Row in Table 6 | Baseline Model Architecture | Training Data           | Enhanced Representation Strategies | Purpose of Comparison                                        |
> | -------------- | --------------------------- | ----------------------- | ---------------------------------- | ------------------------------------------------------------ |
> | 1st row        | InstructMol                 | PubChem Captions (301K) | ✘                                  | Baseline using standard PubChem dataset and atom-level representation |
> | 6th row        | InstructMol                 | KnowMol-100K            | ✘                                  | Shows improvement brought solely by using KnowMol-100K       |
> | Last row       | InstructMol                 | KnowMol-100K            | ✔                                  | Demonstrates further gain from our representation design     |
>
> - **1st vs. 6th row**: Both use the **same model architecture** but differ in the **training dataset**. The performance gain shows the effectiveness of our KnowMol-100K dataset.
> - **6th vs. last row**: Both are **trained on KnowMol-100K**, but only the last row uses our **enhanced representation Strategies**. This highlights the benefit of our enhanced representation strategies over the original InstructMol framework.
>
>
>
> As for other strong baseline models, UniMoT and HIGHT,  **these models do not provide open-source codes, pre-trained checkpoints, or training data**. In the absence of these resources, a rigorous and fair fine-tuning comparison on KnowMol-100K is currently not feasible. We welcome future efforts that open-source these models to enable broader comparative analysis.
>
>
>
> > Q3: The paper does not seem to report the complexity of model training. Since the KnowMol-100K dataset introduces more molecular descriptions, the authors should accurately explain the impact this has on model deployment.
>
> R3: Thank you for raising the important problem.  To clarify, the KnowMol-100K dataset **does not introduce more molecular descriptions in terms of quantity**. Compared to the widely used PubChem Captions dataset (>300K molecules), KnowMol-100K contains only 100K samples. However, our dataset provides **significantly more informative and fine-grained annotations** (covering atomic, functional, structural, and physicochemical levels). PubChem captions suffer from coarse granularity and limited coverage of key molecular aspects. In contrast, KnowMol-100K focuses on **high-quality descriptions** rather than quantity, enabling more effective pretraining.
>
> Regarding model deployment, our method ensures efficiency in two aspects:
>
> (i)Training complexity is substantially reduced. As detailed in **Appendix F**, we provide a comprehensive comparison of training costs between our KnowMol and UniMoT (the second-best model). KnowMol achieves state-of-the-art performance with **only 5 epochs of LLM fine-tuning on 200K samples**, using the **same LLM backbone size** as UniMoT. In contrast, UniMoT requires **10 epochs of LLM fine-tuning on a dataset exceeding 300K samples**, and also involves **two additional pretraining stages** for its molecule encoder, each requiring **50 epochs** on similarly sized >300K large datasets. This demonstrates that our **KnowMol-100K dataset enables more efficient and effective learning**, substantially reducing training complexity with better performance.
>
> (ii) **Additional inference complexity is not introduced**.  Both KnowMol and UniMoT use the same LLM backbone size, and the graph encoder remains frozen during inference. As a result, **the testing-time complexity and latency remain comparable** to the UniMoT.
>
> Due to differences in hardware environments and the lack of released code or training data from UniMoT, we are unable to perform a direct and fair comparison of training time. We advocate for a more open and transparent research community to facilitate reproducible evaluation.

---

> > ### Comment · Reviewer_Cj73 · 2025-08-02
> >
> > Thanks to the author's explanation, this article is now much clearer to me. As for the ablation experiments in Table 6, I hope the author can further improve the analysis in the revised version. In view of the author's resolution of my concerns, I decided to raise my rating to 4.

---

> > > ### Author Response · Authors · 2025-08-02
> > >
> > > We thank the reviewer for the positive feedback and are glad to hear that our clarifications have improved the clarity of the manuscript. Regarding the analysis of the ablation experiments in Table 6, we appreciate your suggestion and will provide a more detailed and insightful discussion in the revised version to strengthen the presentation of the results.

---

### Official Review · Reviewer_YT4S · 2025-07-01

**Rating:** 6
**Confidence:** 4

**Summary:**

This paper focuses on molecular large language models (LLMs). To address the limitations of insufficient textual descriptions and suboptimal molecular representation strategies during pretraining, the authors present a large-scale dataset named KnowMol-100K, which comprises 100K fine-grained molecular annotations across multiple levels. In addition, the paper proposes a chemically informative molecular representation that effectively overcomes the shortcomings of existing molecular encoding methods. Extensive experiments are conducted to evaluate the effectiveness of the proposed KnowMol framework.

**Additional Feedback:**

I have no concerns about this paper. It is well-written and technically sound.

**Dataset Code Accessibility:**

Yes

**Dataset Code Comments:**

The dataset code is available.

**Ethical Considerations:**

No, there are no or only very minor ethics concerns

**Final Justification:**

Thank you for your response. My concerns have been well addressed.

Therefore, I decided to raise my score to 6.

**Limitations Weaknesses:**

1. The proposed dataset is built using GPT-4o to generate the annotations. What about the token cost?

2. Some typos. References [23] and [24] are identical.

3. The proposed KnowMol-100K includes molecular representations in both 1D and 2D modalities. However, it does not consider the 3D modality.

**Strengths Contributions:**

1. This paper presents a large-scale dataset, KnowMol-100K, which consists of 100K detailed multi-level molecular descriptions.

2. This paper proposes a chemically informative molecular representation strategy that combines 1D and 2D representations.

3. This paper proposes a molecular LLM named KnowMol, which outperforms existing models on various molecular understanding and generation tasks.

---

> ### Author Rebuttal · Authors · 2025-07-30
>
> ####
>
> > Q1: The proposed dataset is built using GPT-4o to generate the annotations. What about the token cost?
>
> R1:  Thanks for your question. The KnowMol-100K dataset generation involved approximately 203.5 million input tokens and 44.3 million output tokens, processed via GPT-4o.
>
>
>
> > Q2: Some typos. References [23] and [24] are identical.
>
> R2:  We appreciate the reviewer’s careful reading. We have corrected the issue where references [23] and [24] were inadvertently duplicated; the duplicate has been replaced. In addition, we have thoroughly proofread the manuscript to eliminate other typographical and formatting errors. Thank you for helping us improve the clarity and accuracy of the paper.
>
>
>
> > Q3: The proposed KnowMol-100K includes molecular representations in both 1D and 2D modalities. However, it does not consider the 3D modality.
>
> R3:  We sincerely thank the reviewer for highlighting the limitation regarding the absence of 3D conformational information in our current approach. Indeed, 3D structures are essential for accurately modeling stereochemistry and spatial interactions such as molecular docking. However, our decision to focus on 1D (SELFIES) and 2D (graph-based) representations was driven by practical and methodological considerations: (i) **Scalability and availability**: High-quality 3D conformation annotations are currently unavailable or incomplete in large-scale molecular datasets, especially in the currently widely adopted downstream tasks,  which poses challenges for reliable model training and testing. (ii) **Computational efficiency**: 3D conformer encoding or generation is resource-intensive, which requires additional computation cost.
>
> More importantly, our work is **orthogonal** to 3D structure-based modeling approaches. The primary contribution of this paper lies in two aspects: (i) the construction of KnowMol-100K; and (ii) the design of a chemically-informative representation strategy. Our framework is modular by design and fully compatible with 3D-aware components such as equivariant GNNs or 3D embeddings. Integration of such modules is straightforward once High-quality 3D molecule conformations datasets are available. We view this as a natural and promising extension of our current work and an important direction for future development.

---

> > ### Comment · Reviewer_YT4S · 2025-08-04
> > **Response to Rebuttal**
> >
> > Thank you for your response. My concerns have been well addressed.
> >
> > Therefore, I decided to raise my score to 6.

---

> > > ### Author Response · Authors · 2025-08-04
> > >
> > > We sincerely thank the reviewer for the positive evaluation and are pleased to hear that our responses have fully addressed your concerns! We greatly appreciate your recognition of our work!

---

### Official Review · Reviewer_4z9B · 2025-07-03

**Rating:** 5
**Confidence:** 4

**Summary:**

The paper presents a new dataset comprising 100,000 fine-grained molecular annotations across multiple levels for molecule language model pretraining. Compared to the previous database, the proposed KnowMol addresses the problem that PubChem only covers part of the information essential to train the molecule LLMs. To construct such a dataset, the paper uses RDKit, GPT-4o to annotate molecules. The paper also provides a multi-level function-aware LLM for tokenization. The paper compares the proposed model against multiple baselines in the molecule caption and molecule generation tasks. The paper also includes an ablation study of the proposed framework.

**Additional Feedback:**

Minor comments:

298 Table5-> Table 5

**Dataset Code Accessibility:**

Yes

**Dataset Code Comments:**

The paper provides code and an HF dataset.

**Ethical Considerations:**

No, there are no or only very minor ethics concerns

**Final Justification:**

The authors' response have addressed my questions. Therefore, I raised my score to 5.

**Limitations Weaknesses:**

1. The dataset seems to be generated by GPT-4o. Why does the paper choose GPT-4o instead of some pretrained chemical LLM? If GPT-4o has high accuracy, as the human evaluation suggests, why not include it in the following experiment as a baseline?
2. The idea of  Hierarchical Tokenization is not entirely new. For example, the HEIGHT paper, which is also cited by this paper, already uses graph to generate the tokenization. I would suggest adding another baseline, which uses the HEIGHT architecture but is pretrained on KnowMol-100k to show the effectiveness of newly proposed framework.
3. Some parts of the code are unclear. For example, why is Llava included in GitHub?

**Strengths Contributions:**

1. The proposed new pretraining dataset is a very useful resource, since it covers different levels of molecule structure properties in the instruction tuning data. The paper also evaluates the quality of the proposed framework by chemists.
2. The experiment part seems to be comprehensive. The paper also includes an ablation study for each component.
3. The paper provides both the code and the dataset for readers to reproduce. The figure and table are clear and easy to understand.

---

> ### Author Rebuttal · Authors · 2025-07-30
>
> ####
>
> > Q1: The dataset seems to be generated by GPT-4o. Why does the paper choose GPT-4o instead of some pretrained chemical LLM? If GPT-4o has high accuracy, as the human evaluation suggests, why not include it in the following experiment as a baseline?
>
> R1: We employed GPT-4o during the construction of the KnowMol-100K dataset due to its capability on **zero-shot complex instruction following and multimodal inputs processing**, especially the molecular images in our construction pipeline, which current pretrained chemical LLMs (e.g., MoT5, MolFM, Mol-Instructions) are not able to handle.
>
> Besides, we also made a comparison among GPT-4V, GPT-4o, o1, and GPT-4o-mini for the annotation LLM choosing. The GPT-4o offered the best balance between descriptive accuracy and computational cost for the construction of KnowMol-100K.
>
> The reason we do not include GPT-4o as a baseline is mainly based on **Reproducibility and Fairness**: GPT-4o is a proprietary model with closed weights, no support for fine-tuning, and no transparency about training data. It is difficult to reproduce or fairly compare against open-source models like KnowMol, which is fully trainable and domain-adaptable, and with a guarantee of no test data leakage.
>
>
>
> > Q2: The idea of Hierarchical Tokenization is not entirely new. For example, the HEIGHT paper, which is also cited by this paper, already uses graph to generate the tokenization. I would suggest adding another baseline, which uses the HEIGHT architecture but is pretrained on KnowMol-100k to show the effectiveness of newly proposed framework.
>
> R2: We thank the reviewer for the insightful suggestion. While our work draws inspiration from HIGHT, our **hierarchical tokenization strategy is technically distinct**. HIGHT introduces a hierarchical view primarily using complicated models, such as Vector Quantized Variational Auto Encoders (VQVAEs) on augmented hierarchical graphs, necessitating an additional pretraining cost and significantly increasing computational complexity. In contrast, our method efficiently constructs hierarchical tokens directly from the original molecular graph using the deterministic toolKit RDKit and BRICS algorithm, without bringing additional training parameters or extra model usage in the encoder.
>
> Regarding the suggested baseline—pretraining HIGHT on KnowMol-100K—we acknowledge that this could serve as a valuable comparison. However, **HIGHT has not released source code, model weights, or detailed preprocessing pipelines**. Given the HIGHT architecture's reliance on specific augmentation strategies and hierarchical encodings, it is difficult to make an exact replication.
>
> To rigorously validate the effectiveness of our proposed hierarchical representation framework, we conducted **comprehensive ablation studies** (Table 6). The comparison between rows 6 and 7 demonstrates that removing hierarchical representation leads to substantial performance drops across multiple metrics, underscoring the effectiveness of our hierarchical design beyond conventional atomic-level representations.
>
> Additionally, we appreciate the reviewer's interest in adding the baseline. As an alternative, we add an additional experiment on the forward reaction prediction task using the **PubChem Caption dataset as pretraining data and the hierarchical representation (HR)** to further isolate the contribution of hierarchical representation in the absence of KnowMol-100K.
>
> | Training Data    | Model                       | EXACT↑ | BLEU↑ | LEVENSHTEIN↓ | RDK FTS↑ | MACCS FTS↑ | MORGAN FTS↑ | VALIDITY↑ |
> | ---------------- | --------------------------- | ------ | ----- | ------------ | -------- | ---------- | ----------- | --------- |
> | PubChem Captions | HIGHT-GS                    | 0.293  | 0.935 | 16.687       | 0.774    | 0.618      | 0.566       | 1.000     |
> | PubChem Captions | hierarchical representation | 0.571  | 0.968 | 9.93         | 0.806    | 0.901      | 0.774       | 1.000     |
> | MRQA + DGMG      | hierarchical representation | 0.728  | 0.985 | 6.429        | 0.879    | 0.936      | 0.857       | 1.000     |
>
> As shown in the table, when trained on the same PubChem Captions dataset, **our model with hierarchical representation significantly outperforms HIGHT-GS** across all evaluation metrics. Furthermore, applying **hierarchical representation on KnowMol-100K (MRQA + DGMG)** yields even stronger performance.  The results validate the effectiveness of our hierarchical tokenization approach over HIGHT, even without reliance on better training data.
>
>
>
> > Q3:  Some parts of the code are unclear. For example, why is Llava included in GitHub?
>
> R3: Thank you for pointing this out.  Our code repository was built upon the Llava repository because it provides a well-established code framework with a similar model architecture (i.e., multimodal encoder + feature projector + language model), which aligns with our baseline model. Leveraging Llava allowed us to focus on the dataset training and chemically informative molecular representation learning components without reinventing the core framework. To improve clarity and reproducibility, we will restructure the repository to better distinguish our contributions from the original Llava codebase and add detailed inline documentation to explain each module and modification we introduced.
>
>
>
> > Q4: Minor comments: 298 Table5-> Table 5
>
> R4: We appreciate the reviewer’s careful reading.  We have thoroughly proofread the manuscript to eliminate other typographical and formatting errors. Thank you for helping us improve the clarity and accuracy of the paper.

---

> > ### Comment · Reviewer_4z9B · 2025-08-01
> >
> > Thank you for your rebuttal. The rebuttal has addressed my questions. Therefore, I decided to raise my score to 5.

---

> > > ### Author Response · Authors · 2025-08-01
> > >
> > > We sincerely appreciate your thoughtful consideration of our rebuttal. We are grateful for your constructive feedback and are pleased to see that our clarifications addressed your concerns. Thank you for your support and for recognizing the contribution of our work.

---

### Official Review · Reviewer_rFg5 · 2025-07-19

**Rating:** 4
**Confidence:** 4

**Summary:**

This paper addresses key limitations in existing Molecular Large Language Models (Mol-LLMs), namely the inadequate quality of textual descriptions in pretraining datasets and suboptimal molecular representation strategies. The authors introduce two primary contributions: 1)  KnowMol-100K, a new large-scale dataset of 100,000 molecules with fine-grained, multi-level annotations (atomic, functional group, structural, and physicochemical properties). 2) A set of chemically-informative molecular representation strategies, including the use of SELFIES with a specialized vocabulary for 1D strings and an efficient, parameter-free hierarchical encoder for 2D graphs. Building on these, they develop KnowMol, a multi-modal Mol-LLM that achieves state-of-the-art performance across a comprehensive suite of seven molecular understanding and generation tasks, outperforming existing models.

**Dataset Code Accessibility:**

Yes

**Ethical Considerations:**

No, there are no or only very minor ethics concerns

**Final Justification:**

Thanks for the responses. My concerns have been addressed, and I tend to maintain my scores.

**Limitations Weaknesses:**

- A significant portion of the KnowMol-100K dataset (the structural and physicochemical descriptions) was generated using GPT-4o. While the authors implemented a careful prompting strategy and quality control with human experts, this reliance on a proprietary, black-box model has implications for full reproducibility and potential hidden biases. The expert validation, while valuable, was conducted on a very small subset of 30 samples out of 100,000, which may not be representative of the entire dataset's quality.
- The authors do not report error bars or conduct statistical significance tests for their experimental results, citing the high computational cost of training LLMs. While this is a common issue in the field, it makes it difficult to ascertain whether the observed performance gains, particularly over the strongest baselines like UniMoT where margins can be narrow, are statistically significant.
-The proposed model, like many of its predecessors, relies on 1D (SELFIES) and 2D (graph) molecular representations. While effective, this ignores 3D conformational information, which is crucial for understanding many molecular properties and interactions (e.g., stereochemistry, docking). The authors acknowledge this as a direction for future work in the conclusion, but it remains a limitation of the current approach

**Strengths Contributions:**

- The paper effectively identifies and demonstrates the core weaknesses of current Mol-LLMs. It provides a convincing analysis of the PubChem dataset, highlighting its "imbalanced coverage and coarse granularity" through concrete statistics  and qualitative examples .
- The primary contribution is the KnowMol-100K dataset, which represents a substantial advancement for the field. The construction pipeline is systematic and well-reasoned, combining established tools like RDKit with the advanced capabilities of GPT-4o. The multi-level annotation structure (atomic, functional group, structural, and physicochemical) is comprehensive and provides a much richer textual description than previously available datasets. The authors also conducted a quality inspection with human experts, lending credibility to the dataset's reliability.
- The paper introduces improvements to molecular representation for LLMs: The adoption of SELFIES over SMILES, combined with a dedicated vocabulary, is a well-justified choice to improve robustness and prevent modality confusion. The hierarchical graph encoder is particularly elegant. It captures atomic, functional group, and molecule-level features using a simple pooling strategy on top of a standard GNN, without introducing additional trainable parameters or significant computational overhead. This is an efficient and effective method for providing richer structural information to the LLM.
- authors evaluate KnowMol on an extensive set of 7 tasks, including molecule captioning, property prediction (classification and regression), and four different molecule generation tasks. The model demonstrates state-of-the-art performance, consistently outperforming strong baselines like InstructMol, HIGHT, and UniMoT across the board
- The paper is well-written.

---

> ### Author Rebuttal · Authors · 2025-07-30
>
> ####
>
> > ##### Q1: A significant portion of the KnowMol-100K dataset (the structural and physicochemical descriptions) was generated using GPT-4o. While the authors implemented a careful prompting strategy and quality control with human experts, this reliance on a proprietary, black-box model has implications for full reproducibility and potential hidden biases. The expert validation, while valuable, was conducted on a very small subset of 30 samples out of 100,000, which may not be representative of the entire dataset's quality.
>
> R1: We thank the reviewer for the thoughtful comment.
>
> We fully recognize the importance of reproducibility and transparency in dataset construction, particularly when involving proprietary LLMs, GPT-4o.
>
> To address reproducibility concerns, we have made our **entire data generation pipeline publicly available** and described in detail in Section 2.3, Appendix B, and Figure 3(a), including the basic molecular information from PubChem(e.g., SMILES, molecule image, IUPAC name, etc.), detected functional group annotations from RDKit, all prompt templates,  and example outputs.
>
> The usage of GPT-4o is due to its strong capability in zero-shot complex instruction following and processing multimodal inputs. Researchers could easily replicate the process following our pipeline and using alternative advanced open-sourced LLMs in the future.
>
> Besides, we acknowledge that proprietary models could cause potential hidden biases. To mitigate such risks, GPT-4o's data construction process was carefully designed with multiple safeguards: (i) **Multi-modal Inputs**: Each annotation was conditioned on a combination of structured data sources, including PubChem molecular image, IUPAC names, molecular formulas, SMILES string, and RDKit-derived functional groups. This multi-view information anchors the generation in chemically grounded facts and reduces the influence of any single information or model artifact. （ii) **Template-based Prompting with Expert Knowledge Constraints**:  GPT-4o was prompted using expert-defined, schema-consistent templates that guide its outputs along well-defined semantic constraints, ensuring consistency and controllability across the dataset. (iii)**Diversity-driven Molecule Selection**: To avoid potential bias introduced by narrow molecule diversity, we employed a MaxMin sampling strategy to maximize the molecule's structural diversity across the dataset.
>
> While we acknowledge that no method can completely eliminate model bias, we believe our pipeline substantially reduces the likelihood and impact of such biases on the dataset’s quality and utility.
>
>
>
> Regarding the expert validation, we appreciate the reviewer’s concern about sample size. Our initial quality assessment involved 30 randomly sampled molecules. To strengthen the evaluation, **we conducted an additional independent validation on 30 new randomly sampled entries from the dataset.** All samples were evaluated by the same domain expert using identical criteria.
>
> |                  | Structural Description |                         |                  | Property Description |             |                         | Overall |
> | ---------------- | ---------------------- | ----------------------- | ---------------- | -------------------- | ----------- | ----------------------- | ------- |
> | Factual Accuracy | Completeness           | Clarity and Conciseness | Factual Accuracy | Completeness         | Consistency | Clarity and Conciseness |         |
> | 2.13             | 2.40                   | 2.53                    | 2.60             | 2.20                 | 2.66        | 2.26                    | 2.40    |
>
> The results reaffirmed the high factual accuracy, completeness, and descriptive clarity of the generated annotations. This consistency further confirms the robustness of our data generation pipeline. While we acknowledge that 60 samples still represent a small portion of the full dataset, the **repeated success across independently sampled subsets** provides stronger qualitative evidence of overall quality.
>
> While exhaustive manual evaluation is infeasible, we have already open-sourced our validation scripts in Appendix C and invite the community to contribute additional assessments over time.
>
>
>
> >##### Q2: The authors do not report error bars or conduct statistical significance tests for their experimental results, citing the high computational cost of training LLMs. While this is a common issue in the field, it makes it difficult to ascertain whether the observed performance gains, particularly over the strongest baselines like UniMoT where margins can be narrow, are statistically significant.
>
> R2:  We thank the reviewer for highlighting this critical issue. We fully acknowledge the importance of reporting variance and conducting statistical significance tests, particularly when the observed improvements over strong baselines such as UniMoT are relatively modest.
>
> Due to the high computational cost of training large-scale Mol-LLMs, we follow a common practice in the field (e.g., InstructMol, HIGHT, UniMoT) by reporting single-run results. Nonetheless, to assess the stability of KnowMol, we conducted five independent inference runs on the Molecule Captioning task using the same frozen pretrained KnowMol model (temperature = 1), and report the mean and standard deviation.
>
> Unfortunately, as **UniMoT has not released its code, training data, or model checkpoints**, we were unable to reproduce its results or perform direct statistical significance testing. Therefore, we can only provide a quantitative comparison.
>
> | Model   | BLEU-2↑         | BLEU-4↑         | ROUGE-1↑    | ROUGE-2↑        | ROUGE-L↑        | METEOR↑     |
> | ------- | --------------- | --------------- | ----------- | --------------- | --------------- | ----------- |
> | UniMoT  | 0.664           | 0.583           | **0.722**   | 0.584           | 0.664           | **0.703**   |
> | KnowMol | **0.667±0.002** | **0.595±0.003** | 0.717±0.004 | **0.598±0.002** | **0.670±0.002** | 0.683±0.002 |
>
> The results show that KnowMol outperforms UniMoT on 4 out of 6 metrics, and achieves comparable performance on the remaining ones. The low standard deviations indicate that KnowMol's improvements are stable across inference runs. While formal significance testing is not feasible without access to UniMoT’s codebase, these results suggest that KnowMol's improvements are consistent and not due to random variation.
>
>
>
> > ##### Q3: The proposed model, like many of its predecessors, relies on 1D (SELFIES) and 2D (graph) molecular representations. While effective, this ignores 3D conformational information, which is crucial for understanding many molecular properties and interactions (e.g., stereochemistry, docking). The authors acknowledge this as a direction for future work in the conclusion, but it remains a limitation of the current approach.
>
> R3: We sincerely thank the reviewer for highlighting the limitation regarding the absence of 3D conformational information in our current approach. Indeed, 3D structures are essential for accurately modeling stereochemistry and spatial interactions such as molecular docking. However, our decision to focus on 1D (SELFIES) and 2D (graph-based) representations was driven by practical and methodological considerations: (i) **Scalability and availability**: High-quality 3D conformation annotations are currently unavailable or incomplete in large-scale molecular datasets, especially in the currently widely adopted downstream tasks,  which poses challenges for reliable model training and testing. (ii) **Computational efficiency**: 3D conformer encoding or generation is resource-intensive, which requires additional computation cost.
>
> More importantly, our work is **orthogonal** to 3D structure-based modeling approaches. The primary contribution of this paper lies in two aspects: (i) the construction of KnowMol-100K; and (ii) the design of a chemically-informative representation strategy. Our framework is modular by design and fully compatible with 3D-aware components such as equivariant GNNs or 3D embeddings. Integration of such modules is straightforward once High-quality 3D molecule conformations datasets are available. We view this as a natural and promising extension of our current work and an important direction for future development.

---

### Decision · Program_Chairs · 2025-09-18

**Decision:**

Accept (poster)

**Comment:**

This paper introduces the KnowMol-100K dataset, a valuable resource featuring fine-grained, multi-level molecular annotations. The reviewers unanimously agreed that the dataset and the proposed chemically-informative representation are significant contributions to the field. Initial concerns were raised regarding the reliance on a proprietary LLM for data generation and the lack of statistical significance testing. The authors addressed these points effectively in their rebuttal by providing additional data quality validation and conducting ablation studies to confirm the effectiveness of their approach. The comprehensive and convincing response led reviewers to raise their scores. Overall, the paper is technically sound, well-written, and provides a significant new resource for the community, demonstrating a clear path forward for molecular large language models.